# The evolution of mode-2 internal solitary waves modulated by background shear currents

Peiwen Zhang [1, 3], Zhenhua Xu [1, 2, 3*], Qun Li [4*], Baoshu Yin [1, 2, 3], Yijun Hou[1, 2, 3], Antony K. Liu[5]

[1]Key Laboratory of Ocean Circulation and Waves, Institute of Oceanology, Chinese Academy of Sciences, Qingdao, 266071, China.

[2]Qingdao National Laboratory for Marine Science and Technology, Qingdao, 266071, China.

[3]University of the Chinese Academy of Sciences, Beijing, 100049, China.

[4]Polar Research Institute of China, Shanghai, 200136, China.

[5]Ocean University of China, Qingdao, 266100, China.

*Correspondence to*: Zhenhua Xu (xuzhenhua@qdio.ac.cn); Qun Li (liqun@pric.org.cn)

**Abstract.** The evolution of mode-2 internal solitary waves (ISWs) modulated by background shear currents was investigated numerically. The mode-2 ISW was generated by "lock-release" method, and the background shear current was initialized after the mode-2 ISW became stable. Five sets of experiments were conducted to assess the sensitivity of modulation process to the direction, polarity, magnitude, shear layer thickness and offset extent of background shear current. Three distinctly different shear-induced waves were identified as forward-propagating long wave, oscillating tail and amplitude-modulated wave packet in the presence of shear current. The amplitudes of forward-propagating long wave and amplitude-modulated wave packet are proportional to the magnitude of shear but inversely proportional to the thickness of shear layer, as well as the energy loss of mode-2 ISW during modulation. The oscillating tail and amplitude-modulated wave packet show symmetric variation when the background shear current is offset upward or downward, while the forward-propagating long wave was insensitive to it. For comparison, one control experiment was configured according to the observations of Shroyer et al (2010), in first 30 periods, ~36% of total energy lost at an average rate of 9 W m$^{-1}$ in the presence of shear current, it would deplete the energy of initial mode-2 ISWs in ~4.5 h, corresponding to a propagation distance of ~5 km, which is consistent with in situ data.

**1 Introduction**

Internal solitary waves (ISWs) are commonly observed in stratified oceans, especially in coastal and continental shelf regions (Grimshaw et al., 2010;Helfrich and Melville, 2006; Lamb, 2014). While mode-1 ISWs are frequently observed by in situ observations (Farmer et al., 2009; Klymak and Moum, 2003; Moum et al., 2006) and by remote sensing (Liu et al., 1998; Liu et al., 2004; Zhao et al., 2004; Zhao and Alford, 2006), higher modes are relatively rare captured (Jackson et al., 2013). Even so, with the improvement of observation, mode-2 ISWs have been reported recently (Liu et al., 2013; Shroyer et al., 2010; Yang et al., 2009; Yang et al., 2010). Most of previously reported mode-2 ISWs were categorized as convex types, which have the potential to transport mass (Brandt and Shipley, 2014; Deepwell and Stastna, 2016; Salloum et al., 2012). In contrast, concave mode-2 ISWs are seldom observed because the stratification with a thick middle layer is rare (Yang et al., 2010).

Majority of studies of mode-2 ISWs aimed at interpreting their generation mechanisms under different conditions (Helfrich and Melville, 1986; Huttemann and Hutter, 2001; Stastna and Peltier, 2005; Vlasenko et al., 2010). Under most circumstances, mode-2 ISWs show a specific phenomenon of a "short-lived" nature (Ramp et al., 2012; Terletska et al., 2016; Yuan et al., 2018). Ramp et al. (2012) concluded that mode-2 ISWs around the Heng-Chun Ridge would dissipate in 8.9 hours, suggesting mode-2 ISWs are highly dissipative when traveling around rough topographical features. Terletska et al. (2016) showed the decaying of mode-2 ISWs was induced by step-like topography. The forward-propagating long waves, breather-like internal waves (BLIWs) and oscillating tail were generated during the adjustment process. Yuan et al. (2018) observed the existence of a long mode-1 wave ahead of mode-2 ISW during the evolution of mode-2 ISW over variable topography and found that this process cannot be characterized by Korteweg-de Vries (KdV) theory. The authors suggested using the MITgcm model, which can solve all modes to investigate the integrated evolution process of mode-2 ISWs in variable background conditions.

The ephemeral phenomenon of mode-2 ISWs could also be induced by background shear currents. They are more common in the open ocean because they can be induced by baroclinic eddies, baroclinic tides, wind and mode-1 ISWs (Chen et al., 2011; Wang et al., 1991; Xu et al., 2013; Xu et al., 2016; Stastna et al, 2015). The evolution of mode-1 ISWs in background shear current was extensively studied (Lamb, 2010; Grimshaw et al., 2007; Stastna and Lamb, 2002). Lamb (2010) investigated the

energetics of mode-1 ISWs in a background shear current, providing some methods commonly used to calculate the energy under that circumstance. Stastna and Lamb (2002) considered the effects of background current on mode-1 ISWs and discussed the properties of ISWs during the breaking process. In comparison, few works on mode-2 ISW in shear flow have been produced. Vlasenko et al. (2010) observed mode-2 ISW followed by short wavelength mode-1 oscillating tail in the Luzon Strait. Liu et al. (2013) investigated the generation and evolution of mode-2 ISW in the South China Sea and concluded that the more dispersive mode-2 ISW might not propagate, evolve and persist for long time on the shelf.

Their works focused on the shoaling pycnocline and topography, which were suggest to cause ephemeral mode-2 ISWs, but the effects of background shear currents were neglected. Shroyer et al. (2010) was the sole one recording an integrated evolution process of mode-2 ISW and found that leading mode-2 wave quickly deformed and developed a tail of short, small-amplitude mode-1 wave in the presence of background shear current. Therefore, the authors speculated that the background shear currents could produce instabilities and lead to the adjustment of ISWs and they also concluded that the wave-localized turbulence dissipation was comparable with that induced by mode-1 ISWs. Motivated by the results of Shroyer et al. (2010), we numerically investigated the effect of shear currents on evolution process of mode-2 ISWs. To reveal the sensitivity of the evolution of mode-2 ISWs to variable parameters of background shear currents, we introduced five sets of experiments (21 experiments in total) to generalize our research, including the magnitude, thickness of the shear layer, polarity, direction and offset (center of the shear layer relative to the center of the pycnocline) of the background shear current. These conditions are common in the oceans, for example, the internal tidal wave could be symmetric around the pycnocline (Duda et al., 2004), but in the shelf-slope area, the surface-intensified flow driven by wind could deviate for a distance from the center of pycnocline (Van de Boon, 2011).

The remainder of the paper is organized as follows: The numerical model configurations and background conditions are detailed in Sect. 2. In Sect. 3, the modulation process of mode-2 ISW in background shear currents and its sensitivity to varied background shear currents are presented and assessed. The decaying of mode-2 ISWs' energy and the effect of varied background shear current on it are analyzed and summarized in Sect. 4. In Sect. 5, details of mode-2 ISW evolution and characteristics of shear-induced waves are compared and discussed. Then, the results are summarized in Sect. 6.

## 2 Numerical model and background condition

### 2.1 Model configuration

Our numerical simulations are based on the Massachusetts Institute of Technology general circulation model (MITgcm, Marshall et al., 1997). The nonhydrostatic capability of the MITgcm is turned on because ISWs represent a balance of nonlinearities and dispersions, with the latter derived from nonhydrostatic pressure.

A series of 2D numerical simulations were performed to investigate the evolution of mode-2 ISWs in the presence of background shear currents. The experimental domain was 12.8 km long and 100 m deep. The horizontal and vertical resolutions were 2.5 m with 5120 grids points and 0.5 m with 200 grid points, respectively. The time step was 0.4s in order to ensure that the Courant-Friedrichs-Lewy (CFL) condition was satisfied and the model was stable. The viscosity parameters were set to $10^{-3}$ $m^2s^{-1}$ for the horizontal viscosity $v_H$ and $10^{-4}$ $m^2 s^{-1}$ for the vertical viscosity $v_v$ in the present study. The flux-limiting advection scheme for the tracers introduces numerical diffusivity which is needed for stability, so the explicit diffusivity was set to zero (Legg and Adcroft, 2003, Legg and Huijts, 2006, Legg and Klymak, 2008).

### 2.2 Stratification and background conditions

The choice of background density stratification in all experiments and the shear current in control experiment (case O5) followed the field observation over the New Jersey Shelf (Shroyer et al., 2010). The background stratification and shear currents adopted the hyperbolic tangent function for smoothing, which can be written as follows:

$$\rho(z) = \rho_0 - \frac{\Delta\rho}{2} tanh\left[\frac{(z+z_0)}{h}\right], \tag{1}$$

where $\rho_0 \equiv (\rho_1 + \rho_2)/2$ is the mean density of the vertical water column and $\Delta\rho \equiv (\rho_2 - \rho_1)$ is the difference in density between the upper layer $\rho_1$ (1022 kg m$^{-3}$) and bottom layer $\rho_2$ (1026 kg m$^{-3}$). $h$ is the pycnocline thickness and $z_0$ is the depth of the pycnocline center. The function of background shear current is given by

$$U(z) = U_0 - \frac{\Delta U}{2} tanh\left[\frac{(z+D_s)}{h_s}\right], \tag{2}$$

where $U_0 \equiv (U_1 + U_2)/2$ is the mean background velocity of water column and $\Delta U \equiv (U_2 - U_1)$ is the difference in background horizontal velocity between the upper layer $U_1$ and bottom layer $U_2$, and

$h_s$ is the thickness of shear layer.

In sensitive experiments, the magnitude of the shear current is denoted by $U_d$ and defined as $|\Delta U|$. This value was varied from 0.5 $U_d$ to 2.5 $U_d$ in sensitivity test. Similarly, the thickness of shear layer was varied from 0.5 $h_s$ to 2.5 $h_s$. In cases D1 and P1, an opposing and polarity-reversal background shear currents were initialized for examination, respectively. We further introduced an asymmetry parameter $\Delta$ to investigate the evolution of mode-2 ISWs in offset background shear current (Carpenter et al., 2010). The asymmetry parameter $\Delta$ is defined as follows:

$$\Delta = \frac{D_s - z_0}{h/2} \tag{3}$$

where $D_s$ denotes the depth of shear centre and $h$ denotes the thickness of pycnocline. $\Delta$ was varied from -2 to 2 (case O1 to case O9) to investigate the evolution of the mode-2 ISW in the offset background shear current. The vertical distributions of the background shear currents, density and buoyancy frequency in control experiment are shown in Fig. 1. Detailed configurations of the numerical simulation in the present study are given in Table 1. In all cases, the Richardson number of background current were estimated to be larger than 0.25, indicating the stable state of background environment.

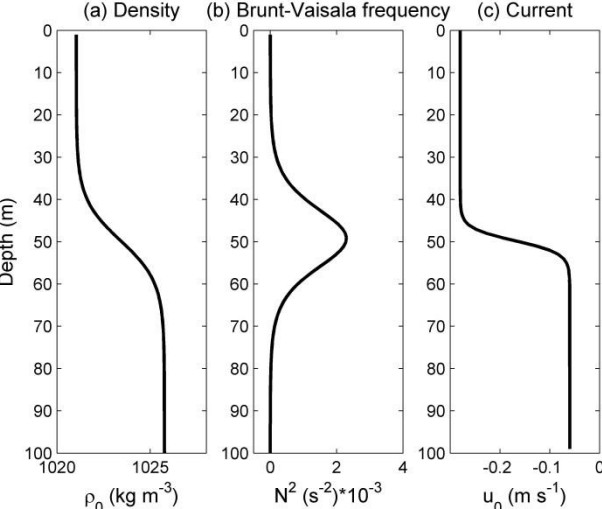

**Figure 1.** Vertical profiles of the (a) density, (b) buoyancy frequency and (c) background velocities in the control experiment (case O5).

**Table 1.** Summary of parameters of variable background shear currents. The depth and thickness of the pycnocline are denoted by $z_0$ and $h$. The thickness of the shear layer is denoted by $h_s$, and the offset cases are indicated by asymmetry parameter $\Delta$. The magnitude of the shear current is denoted by $U_d$, and $U_0$ is the mean background

velocity of the water column. *Por* indicates the polarity of the background shear current, and "+" means a

polarity-reversal shear current. The orientation of the background shear current is indicated by *Ori*, and "+" means

an opposing shear current.

| Case | $z_0$ | $h$ | $h_s$ | $\Delta$ | $U_0$ | $U_d$ | *Por* | *Ori* |
|------|-------|-----|-------|----------|-------|-------|-------|-------|
| **O1** | 50 | 10 | 3 | -2 | 0.17 | 0.22 | (-) | (-) |
| **O2** | 50 | 10 | 3 | -1.5 | 0.17 | 0.22 | (-) | (-) |
| **O3** | 50 | 10 | 3 | -1 | 0.17 | 0.22 | (-) | (-) |
| **O4** | 50 | 10 | 3 | -1.5 | 0.17 | 0.22 | (-) | (-) |
| **O5** | 50 | 10 | 3 | 0 | 0.17 | 0.22 | (-) | (-) |
| **O6** | 50 | 10 | 3 | 0.5 | 0.17 | 0.22 | (-) | (-) |
| **O7** | 50 | 10 | 3 | 1 | 0.17 | 0.22 | (-) | (-) |
| **O8** | 50 | 10 | 3 | 1.5 | 0.17 | 0.22 | (-) | (-) |
| **O9** | 50 | 10 | 3 | 2 | 0.17 | 0.22 | (-) | (-) |
| **H1** | 50 | 10 | 1.5 | 0 | 0.17 | 0.22 | (-) | (-) |
| **H2 (O5)** | 50 | 10 | 3 | 0 | 0.17 | 0.22 | (-) | (-) |
| **H3** | 50 | 10 | 4.5 | 0 | 0.17 | 0.22 | (-) | (-) |
| **H4** | 50 | 10 | 6 | 0 | 0.17 | 0.22 | (-) | (-) |
| **H5** | 50 | 10 | 7.5 | 0 | 0.17 | 0.22 | (-) | (-) |
| **U1** | 50 | 10 | 3 | 0 | 0.225 | 0.11 | (-) | (-) |
| **U2 (O5)** | 50 | 10 | 3 | 0 | 0.17 | 0.22 | (-) | (-) |
| **U3** | 50 | 10 | 3 | 0 | 0.115 | 0.33 | (-) | (-) |
| **U4** | 50 | 10 | 3 | 0 | 0.06 | 0.44 | (-) | (-) |
| **U5** | 50 | 10 | 3 | 0 | 0.005 | 0.55 | (-) | (-) |
| **D1** | 50 | 10 | 3 | 0 | -0.17 | 0.22 | (-) | (+) |
| **P1** | 50 | 10 | 3 | 0 | 0.17 | 0.22 | (+) | (-) |

**2.3 Modal initialization**

5    A rank-ordered mode-2 ISW train was generated by the "lock-release" method without background

current. (Brandt and Shipley, 2014; Olsthoorn et al., 2013; Deepwell and Stastna 2016; Stastna et al.,

2015). Figure 2 demonstrate a schematic diagram of initialization process and the configuration of model parameters. A mixed region was set to be symmetric around the centerline of the pycnocline at the right end, and its length $l_{mix}$ and height $h_{mix}$ were 375 m and 25 m, respectively. The pycnocline was 10 m thick, and this dimension was indicated by $h$. the model was initialized at t = 0 s, and the

5    rank-ordered mode-2 wave train emerged at t = 4000 s and propagated to the left of the domain, as shown in Fig. 2, the leading mode-2 wave is extracted at that time and then propagated for 2000 s until it stabilized. The amplitude $A$ of mode-2 ISWs was defined as the maximum displacement of the upper and lower isopycnals, which are equal in the initial state (Terletska et al., 2016). The wavelength $L$ was defined as the width of the wave at half of the amplitude of the mode-2 ISW in the initial state. At 6000

10   s after the initialization of numerical model, the velocity field of background shear current was superimposed on the model.

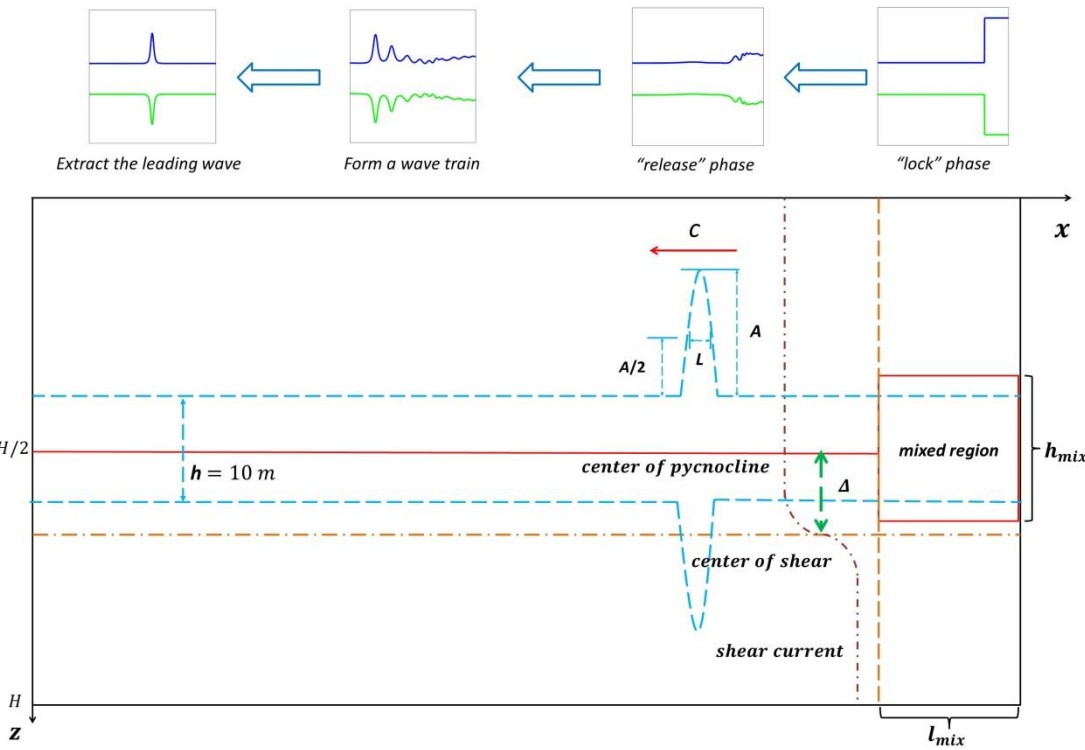

**Figure 2.** Schematic diagram of the modal configuration and initialization, where $\Delta$ denotes the asymmetry

15   parameter, $h$ denotes the thickness of pycnocline, $c$, $A$, and $L$ indicated the propagation speed, amplitude and the wavelength of mode-2 ISW, $h_{mix}$ and $l_{mix}$ denote the height and length of mixed region, and $H$ indicates the depth.

## 3 Results

### 3.1 Characteristics of mode-2 ISWs

The characteristics of initial mode-2 ISW were compared with KdV theory (Grimshaw et al., 2010). The vorticity and density fields of the leading single wave of a mode 2 ISW in the absence of the background shear current are shown in Fig. 3, and the wave exhibits a notably symmetric structure. The ISW had amplitude of 7 m, wave length of 62.5 m with a propagation speed at ~0.31 m s$^{-1}$, which was defined as $c$. It is slightly larger than the linear long-wave phase speed $c_p$ (0.295 m s$^{-1}$) calculated by Taylor-Goldstein equation (Vlasenko et al., 2010) and nondimensionalized by long-wave phase speed as 1.05. The typical time scale $T$ for mode-2 ISW is 200 s which was calculated by $L/c$. The modeled profile of the mode-2 wave is consistent with the theoretical solution of mode-2 ISW in the framework of KdV equation (white line in Fig. 3).

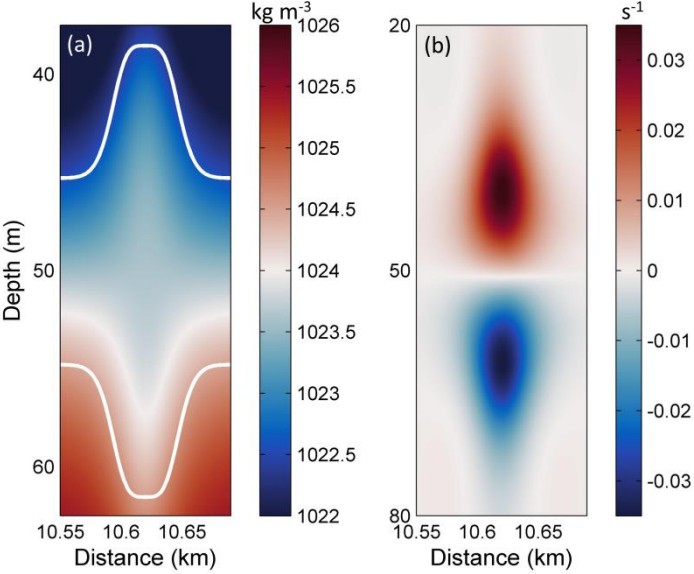

**Figure 3.** The characteristics of a single mode-2 ISW's (a) density field and (b) vorticity in the absence of background shear current. The white lines in (a) demonstrate the theoretical solution of mode-2 ISW in KdV framework.

### 3.2 The evolution of mode-2 ISW in control experiment

The evolution of a mode-2 ISW modulated by background shear current in control experiment (case O5) is shown in Fig. 4, with its corresponding vorticity field shown in Fig. 5. In the initial state (0 T), the mode-2 ISW was symmetric about the pycnocline center and the vorticity of the upper and lower parts counterbalance each other, demonstrating a dipole structure (Fig. 4 (a) and Fig. 5 (a)).

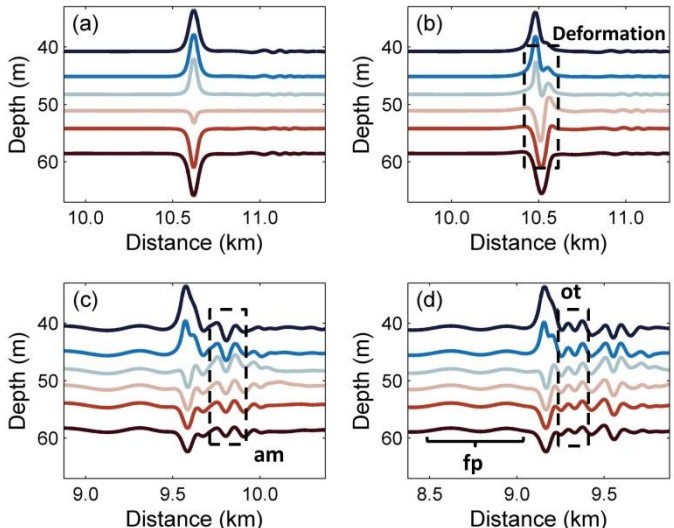

**Figure 4.** The evolution process of the mode-2 ISW in the case O5 (Δ = 0) for the different times (a) 0T, (b) 1.2T, (c) 10T, (d) 14T, where 'am', 'fp' and 'ot' denote the amplitude-modulated wave packet, forward-propagating long wave and oscillating tail, respectively.

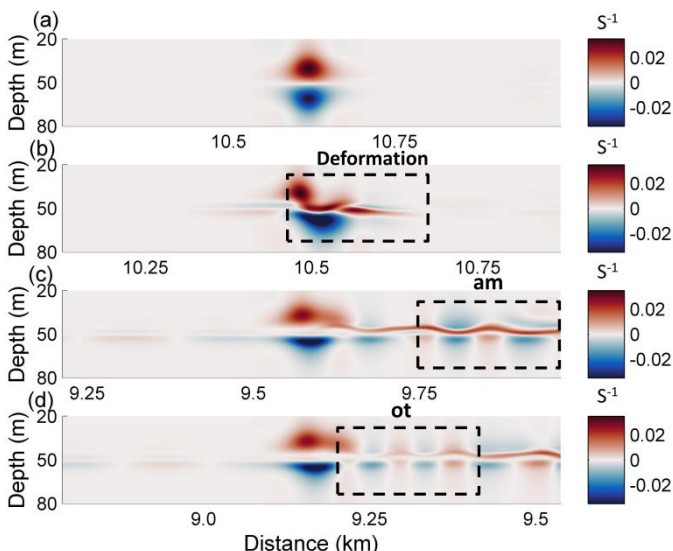

**Figure 5.** The evolution process of the vorticity field (s⁻¹) in the case O5 (Δ = 0) for the different times (a) 0T, (b)1.2T, (c) 10T, (d) 14T, where 'am' and 'ot' denote the amplitude-modulated wave packet and oscillating tail, respectively.

Then, at 1.2 T after the presence of background shear current, as shown in Fig. 5 (b), the shear led to the deformation of the dipole, with the upper part being pushed forward. It also caused the asymmetrical distribution of the vorticity in the horizontal, which is associated with the generation of forward-propagating long waves and the amplitude-modulated wave packet (Fig. 4 (b)), and the latter was defined as a pulsating wave packet (Clarke et al., 2000). The pulsating wave packet propagated

with a steady-state envelope, inside which the waves oscillate freely with different amplitudes (Terletska et al., 2016). To the aft of the ISW, the shear induced by background currents lead to the deformation of the vortex dipole, and an increasing complexity of the vorticity field implied an intensive adjustment occurred. As illustrated in Fig. 5 (c), the vorticity of the mode-2 ISW is redistributed to adapt to the background condition at 10 T. In this process, the vortex of the ISW shrank with the generation of an amplitude-modulated wave packet and a forward-propagating long wave. The amplitudes of shear-induced waves were defined as maximum isopycnal displacement (Stastna and Lamb, 2002). The forward-propagating long wave and amplitude-modulated wave packet can be seen in Fig. 4 (c) with amplitudes of 0.25 m and 1.8 m respectively, and the latter was clearly observed at the rear of the mode-2 ISW.

The oscillating tail caused by shear was visible between the mode-2 ISW and the amplitude-modulated wave packet (Fig. 4 (d)) at 14 T, and it was a radiated mode-1 oscillatory disturbance trailing mode-2 ISW (Stamp and Jacka, 1995). A Hovmöller plot (Fig. 6) of horizontal velocity without the background shear current at the surface was plotted. The forward-propagating long wave, oscillating tail and amplitude-modulated wave packet were found to propagate persistently. The amplitude-modulated wave packet propagated independently from the ISW, indicating that it generated transiently. Based on the vorticity field (Fig. 5 (d)) and the time-space varying nature (Fig. 6), the generation of the oscillating tail and the forward-propagating long wave were continuously sustained by the energy of the ISW with decreasing rate. Therefore, they have the potential to drain the energy from an ISW over a long time scale. The traces of forward-propagating long wave and amplitude-modulated wave packet appear at the same time (Fig. 6). The oscillating tail appears later than the amplitude-modulated wave packet, and is located between the mode-2 ISW and amplitude-modulated wave packet (Fig. 6). Thus, the energy loss of the ISW caused by forward-propagating long waves occurs earlier than oscillating tail.

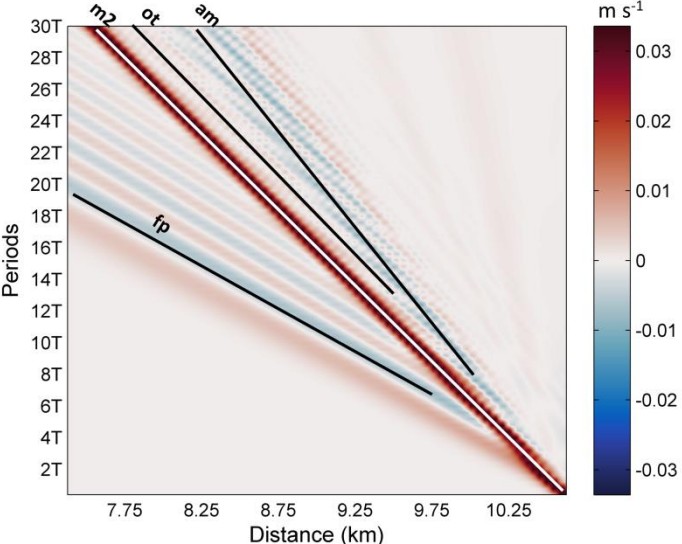

**Figure 6.** Hovmöller plot of horizontal velocity without background shear current at the surface. The mode-2 ISW, forward-propagating long wave, oscillating tail and amplitude-modulated wave packet are denoted by 'm2', 'fp', 'ot' and 'am', respectively.

### 3.3 The evolution of mode-2 ISW in offset background shear current

The influence of the offset background shear current on the modulation of mode-2 ISW was investigated, and in offset cases the shear current was set to deviate from the center of pycnocline with varied asymmetry parameters. We take case O9 ($\Delta = 2$) for a detailed examination in the following section. The evolution of the mode-2 ISW in the case O9 and its corresponding vorticity field are

10 shown in Fig. 7 and Fig. 8, respectively.

**Figure 7:** The evolution processes of the mode-2 ISW in the case O9 ($\Delta = 2.0$) for the different times (a) 0T, (b) 1.2T, (c) 10T, (d) 14T, where 'am', 'fp' and 'ot' denote the amplitude-modulated wave packet, forward-propagating long wave and oscillating tail, respectively.

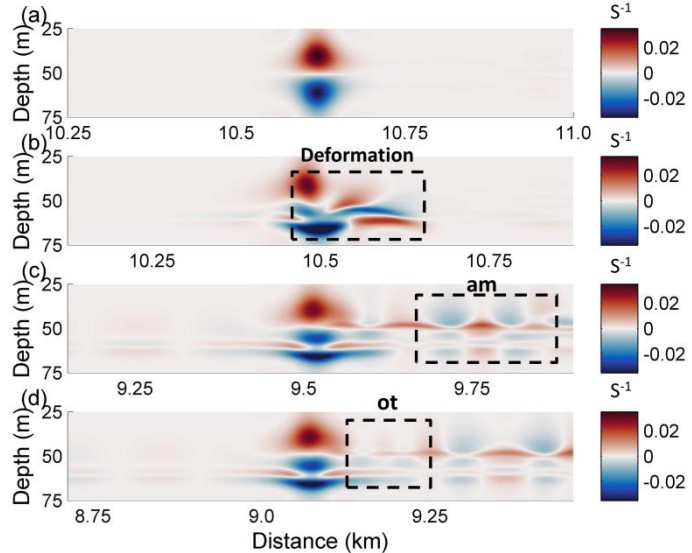

**Figure 8.** The evolution process of the vorticity field in the case O9 ($\Delta = 0$) for the different times (a) 0T, (b)1.2T, (c) 10T, (d) 14T, where 'am' and 'ot' denote the amplitude-modulated wave packet and oscillating tail, respectively.

The deformation of the vortex dipole at 1.2 T after the initialization of background shear current is illustrated in Fig. 8 (b). The shear vertically distorted the lower section of the vortex and forced some of the vortices from the lower part of the dipole to penetrate the upper part. Redistribution of vorticity occurred both vertically and horizontally. The upper section was bifurcated such that the branch containing most of the vorticity intruded ahead, which related to the generation of a forward-propagating long wave in the same manners at that in the control experiment, as shown in Fig. 7 (b). The other branch moved backward to the aft of the ISW, corresponding to the generation of amplitude-modulated wave packet. Given this modulation process, while the lower part of the dipole obviously shrank, the upper parts slowly reunited to restore the vertical balance.

The vorticity distribution was different from that of the control experiment at 10 T (Fig. 8 (c)). The lower part of the vortex had two obvious cores, and they were separated by shear effect and jointly balanced with the upper section. The amplitude-modulated wave packet and forward-propagating long waves are plotted (Fig. 7 (c)). The amplitude of the wave packet was 1 m, which was smaller than that

of the control experiment. However, the amplitude of the forward-propagating long wave was still approximately 0.2 m. In Fig. 7 (d), an oscillating tail with 0.25 m amplitude developed at the rear of the wave. It was sustained by the energy transferred from the ISW, as shown in Fig. 8 (d). When the shear current is offset in the upward direction ($\Delta < 0$), the asymmetry of the mode-2 ISW during the modulation became clearer. The amplitude of the oscillating tail and amplitude-modulated wave packet both decreased when the shear current was offset upward, showing a symmetric variation trend with respect to the downward offset condition. For the forward-propagating long wave, its amplitude oscillates by approximately 0.2 m, suggesting the insensitive nature of the long wave to the upward offset shear current. The small amplitude of the oscillating tail and amplitude-modulated wave packet in larger $\Delta$ cases indicate that the modulation could be weaker when $\Delta$ increased, and the weakening of the oscillating tail makes the amplitude-modulated wave packet more visible.

### 3.4 The evolution of mode-2 ISW in opposing and polarity-reversal background shear current

The modulation of mode-2 ISW in following (control experiment) and opposing shear current was compared. In case D1, the background shear current oriented against the mode-2 ISW. The general pattern of the forward-propagating long wave, oscillating tail and amplitude-modulated wave packet were similar to those in the control experiment. In this opposing case D1, the amplitude of the forward-propagating long wave and oscillating tail were not significantly affected. For the mode-2 ISW, its amplitude also remains nearly unchanged between the following and opposing cases, while the amplitude-modulated wave packet's amplitude decreased from 1.85 m (following case) to 1.09 m in the opposing case. These results suggest that only the amplitude-modulated wave packet is sensitive to the orientation of the background shear current. The modulation of mode-2 ISW in polarity-reversal shear current was also compared to the control experiment. In case P1, the polarity-reversal background shear current was initialized in the model. The properties of the wave structures in the case P1 and the control experiment were compared and no significant difference were found. Only the polarity of the forward-propagating long wave, oscillating tail and amplitude-modulated wave packet are reversed in case P1. This result indicates that the polarity of those shear-induced wave structures is closely related to the polarity of the background shear current.

### 3.5 The evolution of mode-2 ISW in background shear current with varied magnitude

The modulations of mode-2 ISWs in variable magnitude of shear currents were characterized. The magnitude of the background shear current was varied from 0.5 $U_d$ to 2.5 $U_d$ from case U1 to case U5 to study its influence on the evolution of the mode-2 ISW. In case U5 (Fig. 9), a relatively larger amplitude forward-propagating long wave was observed at 20 T (Fig. 9 (c)). The mode-2 ISW became inconspicuous at 50 T (Fig. 9 (d)), while an amplitude-modulated wave packet propagated clearly. The increasing magnitude of the background shear current leads to smaller amplitudes of the mode-2 ISW in both upper and lower parts. In the larger magnitude case, the amplitude-modulated wave packet and forward-propagating long wave were significantly strengthened, and their amplitudes reached 3.75 m and 0.38 m (case U5), respectively. In contrast, a larger magnitude didn't make the amplitude of the oscillating tail continue to increase. In the larger magnitude case, the oscillating tail was unable to be clearly observed. Its amplitude becomes smaller (0.25 m) than that of the forward-propagating long wave (0.38 m). In summary, all three shear-induced waves are sensitive to the magnitude of the shear current.

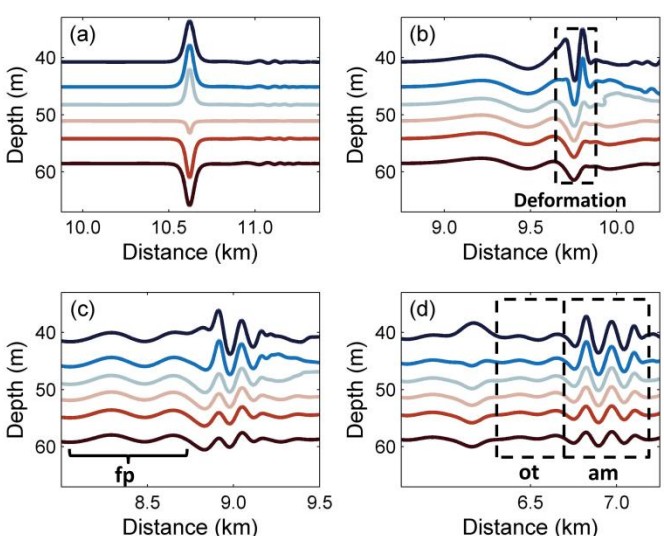

**Figure 9.** The evolution processes of the mode-2 ISW in the case U5 for the different times (a) 0T, (b) 10T, (c) 20T, (d) 50T, where 'am', 'fp' and 'ot' denote the amplitude-modulated wave packet, forward-propagating long wave and oscillating tail, respectively.

**3.6 The evolution of mode-2 ISW in background shear current with varied thickness of shear**

The thickness of the shear layer $h_s$ was also varied to investigate its effect on the modulation of mode-2

ISW (cases H1 to H5). In comparison among these cases, the forward-propagating long wave's amplitude decreased with larger $h_s$, reaching 0.17 m in case H5 with 2.5 $h_s$. The amplitude-modulated wave packet and oscillating tail both decrease to 1.33 m and 0.42 m in amplitude in larger $h_s$ (case H5), respectively, while the mode-2 ISW's amplitude reaches 7.95 m. This result shows that the background shear current with smaller $h_s$ could only moderately deform the mode-2 ISW. As a result, all three shear-induced wave structures are sensitive to the variation in the thickness of the shear layer.

### 3.7 The relationship between the evolution process and variable parameters of background shear current.

The amplitudes of the forward-propagating long wave, oscillating tail and amplitude-modulated wave packet in varied background shear currents are summarized to investigate their sensitivity to the varied background shear currents. The amplitude of the forward-propagating long wave and amplitude-modulated wave packet are proportional to the magnitude of the background shear current, but the oscillating tail is insensitive to the higher magnitude of background shear current (Fig. 10 (a)) . The amplitudes of the oscillating tails and amplitude-modulated wave packet are inversely proportional to the thickness of the shear layer, and the forward-propagating long wave decreased slightly in amplitude with increasing $h_s$ (Fig. 10 (b)). To reveal the effects of the $\Delta$ on those shear-induced wave structures, a comparison of different cases from $\Delta = 0$ (case O5) to $\Delta = 2$ (case O9) was given. The modulation caused by background shear currents was weakened as the $\Delta$ increased, corresponding to the decreased amplitude of the amplitude-modulated wave packet and oscillating tail. The amplitude-modulated wave packet has the highest amplitude among all cases compared to those of the other two wave forms. The amplitude of the amplitude-modulated wave packet decreased from 1.8 to 1 m monotonically, and the amplitudes of the oscillating tails decreased from 0.85 to 0.25 m between the case O5 ($\Delta = 0$) and the case O7 ($\Delta = 1$) but remained stable between the case O7 ($\Delta = 1$) and case O9 ($\Delta = 2$), indicating that the amplitude-modulated wave packet were more sensitive to the $\Delta$ than the oscillating tail. As expected, the ratio between the amplitude of modulated wave packet and oscillating tails increased from 2.1 in the case O5 to 4 in the case O9, so the amplitude-modulated wave packet became more distinct in case O9. In contrast, the forward-propagating long wave barely affected by $\Delta$ and remained constant at approximately 0.2 m in all cases. A similar variation trend could be found in

the upward offset cases (Fig .10 (c)). The amplitudes of the oscillating tail and amplitude-modulated wave packet decreased monotonically as the shear current was offset upward. The forward-propagating long wave was barely affected by Δ and remained constant at approximately 0.2 m in all offset cases. This divergence of sensitivity between the forward-propagating long wave, amplitude-modulated wave packet and oscillating tail could be related to their generation mechanisms, which are discussed in section 5.

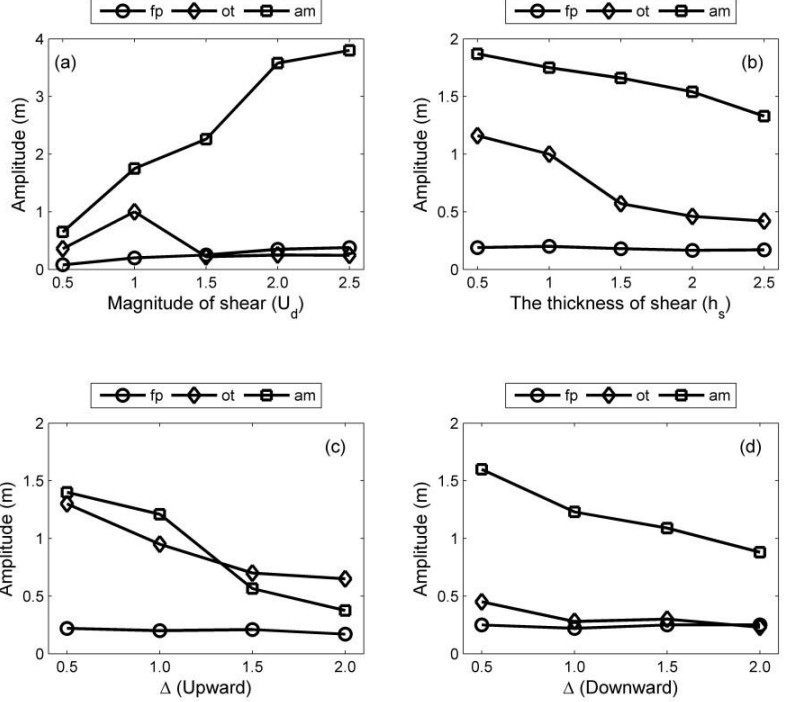

**Figure 10.** The summarized results of the amplitudes of the forward-propagating long wave (denoted by 'fp'), oscillating tail (denoted by 'ot') and amplitude-modulated wave packet (denoted by 'am') with the presence of (a) varied magnitude of shear currents at 40 T, (b) varied thickness of shear currents at 40 T, (c) upward offset background shear currents at 30 T and (d) downward offset background shear currents at 30 T.

## 4 Energy analyses

### 4.1 Calculation of energy

The evolution of the mode-2 ISWs in the presence of shear currents was analyzed quantitatively in terms of energy. The available potential energy (APE) and kinetic energy (KE) for a region were

calculated based on the method suggested by Lamb (2010):

$$KE = \int_{x_l}^{x_r} \int_{-H}^{0} \rho_0 (u^2 + w^2) dx dz, \tag{4}$$

$$APE = \int_{x_l}^{x_r} \int_{-H}^{0} (\rho - \bar{\rho}) g z dx dz, \tag{5}$$

And the total energy E is written as:

$E = KE + APE,$                            (6)

where $\bar{\rho}$ is the reference density extracted from the initial field, $\rho_0$ is the averaged density and $\rho$ is the fluid density. $x_r$ and $x_l$ are the boundary locations of the integration region, and the $x$ satisfies $x_l \leq x \leq x_r$. During the calculation of the wave energy, $x_r$ and $x_l$ denote the left and right boundaries, respectively, where the available potential energy flux equals zero (Lamb, 2010). $u$ and $w$ are the

horizontal and vertical velocity induced by the wave, respectively. The total energy of the initial mode-2 ISW just before the introduction of the shear calculated by the above expressions was 146.2 KJ m$^{-1}$. .

Using the method introduced by Lamb and Nguyen (2009), we set two transects at the front and rear edges of the mode-2 ISW to compute the energy fluxes radiating from the ISW. The total energy flux

through a transect is:

$$E_f = KE_f + APE_f + W, \tag{7}$$

where $KE_f, APE_f$ and $W$ are the kinetic, available potential and pressure perturbation energy fluxes, respectively. They are written as:

$$KE_f = \int_{-H}^{0} u E_{ke} dz, \tag{8}$$

$$APE_f = \int_{-H}^{0} u E_{ape} dz, \tag{9}$$

$$W = \int_{-H}^{0} u p_d dz, \tag{10}$$

where $u$ is the horizontal velocity induced by mode-2 ISWs, $E_{ke}$ and $E_{ape}$ are kinetic energy and available potential energy, and $p_d$ is the pressure perturbation relative to the reference state (Lamb and Nguyen, 2009).

### 4.2 The cascading process of energy

To understand better how the mode-2 ISW is modulated with the presence of shear current and to determine the nature of the whole wave system, the EOF (Empirical Orthogonal Function) method was

applied to the modal decomposition. It is commonly used for mode decomposition and space-time-distributed datasets examination in oceanography (Venayagamoorthy and Fringer, 2007), especially with strong nonlinearity properties, where the traditional normal mode decomposition is not suitable (Venayagamoorthy and Fringer, 2007). Because of the shear effect on the mode-2 ISWs, the energy can cascade into all the modes, including mode-1 and higher modes. The vertical and horizontal kinetic energy modal distributions of different wave forms shed from the mode-2 ISW in the case O5 are shown in Fig. 11 and the model structures on waves (Talipova et al., 2011) for different times are given in Fig .12. In the forward-propagating long waves, the kinetic energy was all in the mode-1 form because of its propagation in front of the ISWs, indicating the generation of the forward-propagating long waves corresponds to a cascading process from higher to lower modes. Fig. 11 shows that the energy was mainly mode-1 in the oscillating tail and the amplitude-modulated wave packet, but weak mode-2 signals were also present. The presence of mode-2 energy for the oscillating tail is reasonable because they have shorter wavelengths and slower phase speeds than the mode-2 ISW and propagate following the mode-2 ISW (Akylas and Grimshaw, 1992; Vlasenko et al., 2010). The model structures of forward-propagating long wave for different times show its mode-1 nature was stable during the evolution of mode-2 ISW in the background shear current (Fig. 12 (a) and (b)). In the rear of the mode-2 ISW, the model structures of trailing waves transformed slightly with the time (Fig. 12 (c) – (g)), the wave-induced current are more and more concentrated around the pycnocline when the amplitude-modulated wave packet propagates far away from the mode-2 ISW.

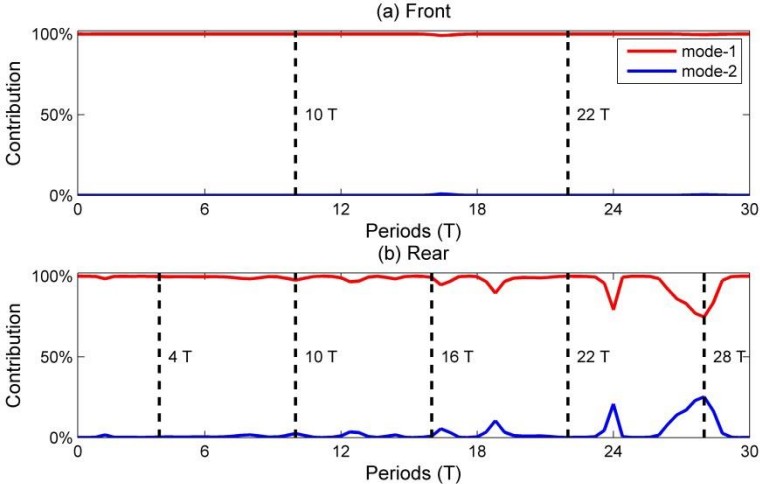

**Figure 11.** Percent contributions of mode-1 and mode-2 to the total kinetic energy in control experiment (case O5 with $\Delta = 0$) from 0 to 30 T at the (a) front and (b) rear of the mode-2 ISW, the dash lines indicate the cross section

where the model structures on waves shown in Figure 12.

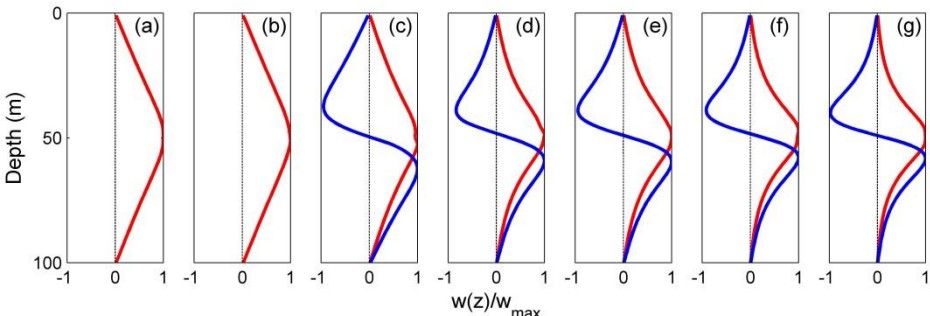

**Figure 12**. The model structures on mode-1 (red solid line) and mode-2 (blue solid line) waves in front of the mode-2 ISW at (a) 10 T, (b) 22 T and in rear of the mode-2 ISW at (c) 4 T, (d) 10 T, (e) 16 T, (f) 22 T and (g) 28 T for control experiment (case O5).

## 4.3 Energy loss of mode-2 ISW

The energy loss of mode-2 ISW in control experiment was investigated in detail to demonstrate the corresponding energy changing process during the modulation and compared with the observations of Shroyer et al. (2010). In this case, ~36% of the total energy of the mode-2 ISW was lost by 30 T, corresponding to a propagation of 1.86 km, part of that energy was transferred to shear-induced wave forms. During the first 30 T, The average energy loss rate was 9 W m$^{-1}$. Modulated by background shear current, the mode-2 ISW exhibits a highly dissipated nature, and the high energy loss rate is comparable to that of the longer mode-1 ISW (Lamb and Farmer, 2011; Shroyer et al., 2010). This quantitative result was consistent with the observation data (Shroyer et al., 2010).

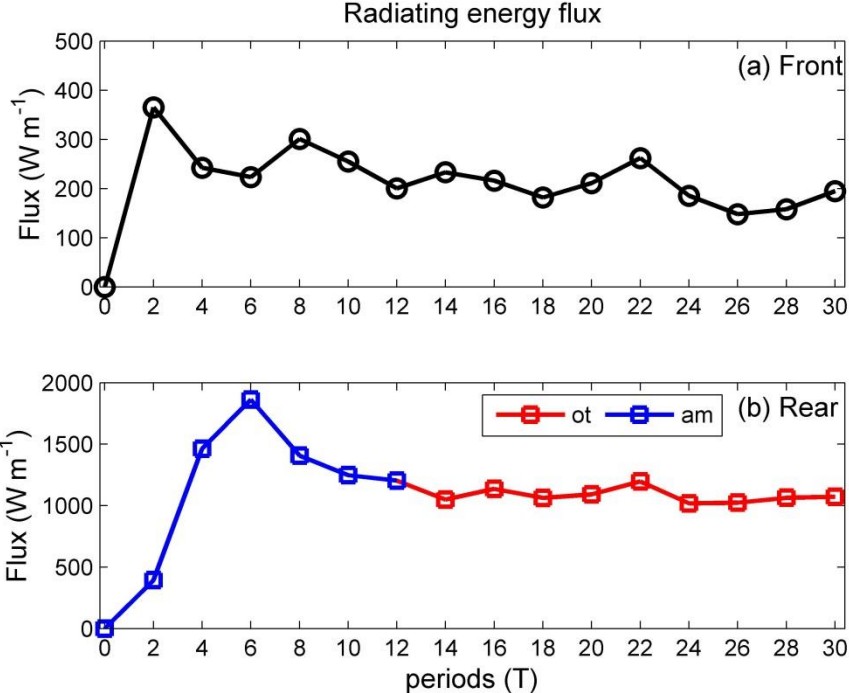

**Figure 13.** Vertical integrals of the radiating energy flux in the (a) front and (b) rear transects of the mode-2 ISW in control experiment (case O5 with $\Delta = 0$) at different times, where "ot" denotes the oscillating tail and "am" denotes the amplitude-modulated wave packet.

We further calculated the radiating energy flux (Fig. 13) to investigate the detailed energy transport. The pressure perturbation generally make the largest instantaneous contributions to the total energy flux (Lamb and Nguyen, 2009; Venayagamoorthy and Fringer, 2007). For an ISW, the pressure perturbation term could be dominant (Lamb 2007). Since we focused on the energy loss of the mode-2

10   ISW, only a total energy flux was analyzed in the following paragraph. The radiating energy flux in the front transect slowly decreased, indicating the forward-propagating long wave drains the energy of the mode-2 ISW at a decreasing rate in the presence of a background shear current. In the rear transect, the radiating energy flux decreased from 1.8 KW m$^{-1}$ to 1.2 KW m$^{-1}$ before stabilizing at approximately 1.0 KW m$^{-1}$ above 12 T. The energy flux at the crest of the amplitude-modulated wave packet and the

15   oscillating tail ranged from 1.0 KW m$^{-1}$ to 2.0 KW m$^{-1}$ (Blue solid line in Fig. 13) and 1.0 KW m$^{-1}$ to 1.1 KW m$^{-1}$, respectively. Combining with the evolution process, the high radiating flux before 12 T indicates the generation process of the amplitude-modulated wave packet and that the relative low radiating energy flux above 12 T is caused by the generation of an oscillating tail.

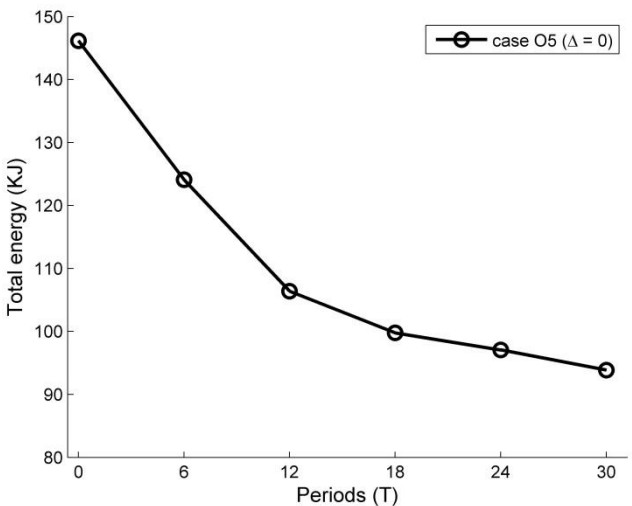

**Figure 14.** The total energy of the ISW at the different times in case O5

The total energy of the mode-2 ISWs at different times in control experiment are shown in Fig. 14. From 0 to 6 T, the averaged energy loss rate was 18.4 W m$^{-1}$. This period corresponded to the generation of the amplitude-modulated wave packet and forward-propagating waves, during which a deformation was caused by shear to the aft of the ISW. The averaged energy loss rates from 6 to 12 T decreased. This stage contained the exit of the amplitude-modulated wave packet, which occurred at approximately 10 T (Fig. 4 and Fig. 7), and the generation of the oscillating tail. In this case, ~34% of the total energy loss (52.34KJ m$^{-1}$) lost at an average rate of 14.8 W m$^{-1}$ from 6 to 12 T, and some of the energy transferred to the amplitude-modulated wave packet. Combining the results of the energy flux in front and rear of the mode-2 ISW (Fig. 13), thus, in the early stage of modulation, the amplitude-modulated wave packet could make larger contribution to the energy transfer process. After the amplitude-modulated wave packet shed from the mode-2 ISWs, sharply decreased loss rates could be seen in 12 – 18 T, during which the energy loss was caused by forward-propagating long waves and the oscillating tail. The shear currents continuously sustained the development of the oscillating tail and the forward-propagating long waves. Thus, these two forms could slowly drain the energy of the ISW, with an average rate of 3.5 W m$^{-1}$. In the following periods, with the forward-propagating long waves and oscillating tails, the ISWs decayed with a relatively low rate. This was reinforced by a similar result given by Olsthroon et al. (2013).

**4.3 The relationship between the energy loss of mode-2 ISW and variable parameters of background shear current**

We summarized the effect of variable parameters of shear current on the energy loss of mode-2 ISW during the modulation (Fig. 15). The polarity and the direction of the background shear current have minor effect on the energy loss of mode-2 ISWs. The energy loss of the mode-2 ISW was proportional to the magnitude of shear current, but inversely proportional to the thickness of shear layer (Fig. 15 (a) and (b)). For case U5, 78.04 KJ m$^{-1}$ energy loss in 30 T, ~53% of total energy of mode-2 ISW and the averaged energy loss rate was 13 Wm$^{-1}$, indicating the magnitude of shear could significantly increase the energy loss of mode-2 ISW. in contrast, for case H5, 42.05 KJ m$^{-1}$ energy loss in 30 T, ~29% of total energy of mode-2 ISW and the averaged energy loss rate was 7W m$^{-1}$, showing that a larger thickness of shear has opposite effect on the energy loss of mode-2 ISW. In the offset background shear currents, the energy loss of mode-2 ISW monotonically decreased with an increasing $\Delta$, showing a symmetric trend in both upward and downward offset cases (Fig. 15 (c) and (d)). Therefore, the energy losses of mode-2 ISWs are sensitive to the magnitude, thickness and offset extent, but insensitive to the polarity and direction of background shear current. We further found that the energy losses of mode-2 ISW in upward conditions were larger compared to the downward conditions. This phenomenon is caused by the asymmetry of wave-induced shear. When the shear layer moves up or down, the background current could weaken the shear in upper layer or strengthen the shear in lower layer, causing the difference in energy losses of mode-2 ISW.

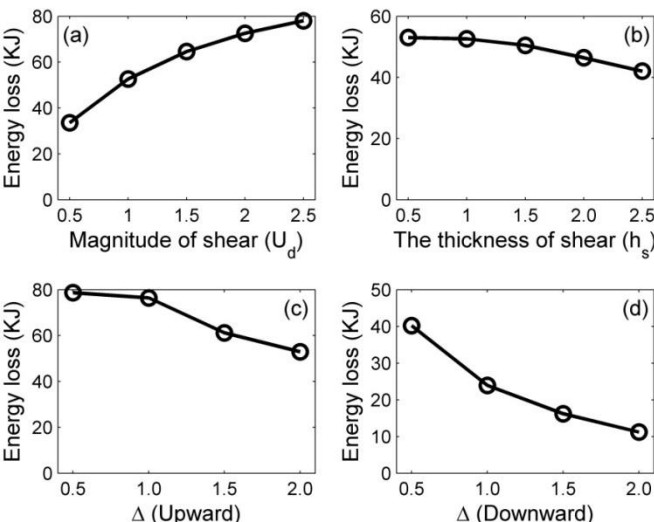

**Figure 15**. The summarized results of the energy loss of the mode-2 ISW at 30 T with the presence of (a) varied magnitude of shear currents, (b) varied thickness of shear currents, (c) upward offset background shear currents and (d) downward offset background shear currents.

**5 Discussions**

We compared our results of the control experiment (case O5) with the observation of Shroyer et al. (2010) for validation. The wavelengths and amplitudes of the initialized mode-2 ISWs were selected to be comparable to the observation of Shroyer et al. (2010) on the New Jersey Shelf. A depression wave

at the rear of the ISW in the first transect of wave *Jasmine* was similar to the wave form around 10 T in the numerical simulation (Fig. 3 (a) in Shroyer et al., 2010). Thus the first, second and third transects of wave Jasmine corresponded to 10 T, 23 T and 38 T in the case O5, respectively. The energy loss rates in 10 T and 23 T were 14.8 W m$^{-1}$ and 2.3 W m$^{-1}$, the averaged energy loss rate between 10 T and 38 T was 4.1 W m$^{-1}$, they were in same scale with the corresponding observations. Between 10 T and 38 T,

~16% of mode-2 ISW total energy lost,it was a little smaller than the typical observation results. Relatively high energy loss rate and a large amplitude oscillating tail in the third transect of the field observations could be probably attributed to the effect of a shoaling pycnocline (Shroyer et al., 2010) since the enhancement of the asymmetries in stratification could increase the energy loss of the wave during the propagation of mode-2 ISWs (Carr et al., 2015; Olsthoorn et al., 2013).

We also compared our results with Stastna et al. (2015), who investigated the mode-2 ISW interaction with mode-1 ISW at the same scale. The authors concluded that the shear current is vital, while the deformation of the pycnocline only slightly altered the structure of the mode-2 ISW. For our results, we focused on the effect of shear currents induced by baroclinic eddies, baroclinic tides or wind. We found a deformation of mode-2 ISW and it illustrated asymmetry during the modulation in the presence of

background shear current, it is coincident with the conclusion given by Stastna et al. (2015).

Then, we further discussed the characteristics of shear-induced waves. In our simulations, the modulation of mode-2 ISW in the presence of shear currents excites the amplitude-modulated wave packet with characteristics of breather-like internal wave (Terletska et al., 2016). Internal breather waves are periodically pulsating, isolated wave forms, they are also a type of steady state wave solution

of the extended Korteweg-de Vries equation (Lamb et al., 2007), and has been found to exist in the real ocean (Vlasenko and Stashchuk, 2015). We introduced the definition of breather by Clarke et al. (2000) to clarify the characteristics of amplitude-modulated wave packet. The envelop lines of the

amplitude-modulated wave packet in case O5 are shown in Fig. 16. Inside the envelop lines, the oscillatory pulses freely oscillate, satisfying the breather definition. Additionally, the energy inside the envelope was calculated and remains nearly constant from 12 to 28 T. Similar to the case O5, the characteristics and energy loss of the amplitude-modulated wave packet for the case O9 were also

5      similar with the breather definition. Similar results were revealed by Terletska et al. (2016), the interaction of mode-2 internal waves with a step-like topography could induce the generation of BLIWs (breather-like internal waves), providing a possibility of the breather generation in a thin intermediate layer with a range of intermediate wavelength.

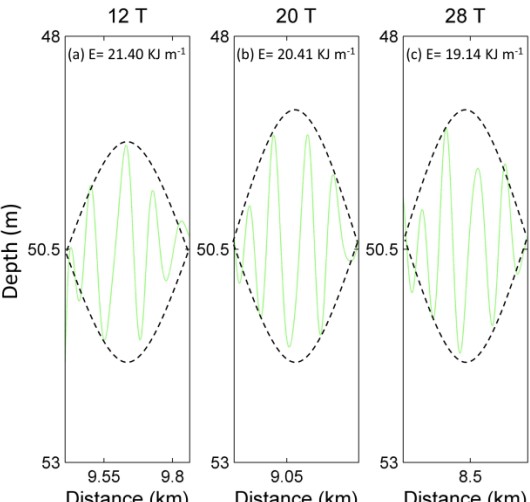

**Figure 16.** The envelopes of the amplitude-modulated wave packet in the case O5 ($\Delta = 0$) at (a) 12 T, (b) 20 T and (c) 28 T, where the mean density isopycnals of upper and lower layer (green line) are plotted.

An oscillating tail induced by shear was also observed in similar studies. The generation of this feature

15     could be related to the shear, and the tail was sustained by continuous energy input. The presence of the background shear current modulated the mode-2 ISW and induced the continuously energy transfer process from the main wave to the oscillating tail, supporting its existence. Forward-propagating long waves were also observed by Yuan et al. (2018), who found that some small but significant long wavelength mode-1 waves appeared ahead of mode-2 ISWs. The forward-propagating long wave was

20     generated by the collapse of mixing induced by shear instability, and it could drain the energy of mode-2 ISWs at a decreasing rate, leading to an inevitable energy loss of those mode-2 ISWs in the presence of background shear. The results in section 3.7 show that the amplitude of the

forward-propagating long wave is proportional to the magnitude of the shear current, indicating that the forward-propagating long wave was affected by the strength of shear. Δ denotes the offset extent of the background shear current, and the strength of the shear remains unchanged when Δ varied. Therefore, the forward-propagating long wave was insensitive to variation in Δ.

The mechanism of adjustment during the modulation has been reviewed. The superposition of an initially stable shear current and the mode-2 ISW induced a low Ri region with a minimum value of less than 0.01 in our simulation, indicating a possible development of shear instability (Barad and Fringer, 2010). The ISW tends to adjust gradually and adapt to the new background conditions. The vorticity and Ri of adjustment process for the mode-2 ISW in the case O5 is shown in Fig. 17. The Ri
values are larger than 0.25 before 0.8 T. After 0.8 T, due to the shear currents and weakened stratification, the lowest Ri values decrease below 0.01 and are accompanied by increased vorticity around the low Ri region, indicating the generation of shear instability (Pawlak and Armi, 1998). The overturning in isopycnal could be also observed in corresponding low Ri region (Fig. 18). Then, the Ri values increased larger than 0.25, and the stratification is restored above 6.8 T. For the case O9 (Fig.
19), before 0.8 T, the Ri values are also larger than 0.25. They decrease below 0.01 near the depths of the shear current after 0.8 T, which is accompanied by increased vorticity and weakened stratification, indicating the occurrence of the shear instability. The stratification is restored and the Ri values increased larger than 0.25 after 2 T. Compared to the case O5, the region with low Ri and increased vorticity was smaller in the case O9, making the instability process was less apparent, and the shear
instability for the case O5 occurs at the same time but lasts longer than that for the case O9. Those comparison illustrated the adjustment of mode-2 ISWs modulated by shear current are more energetic in overlap cases compared to offset cases.

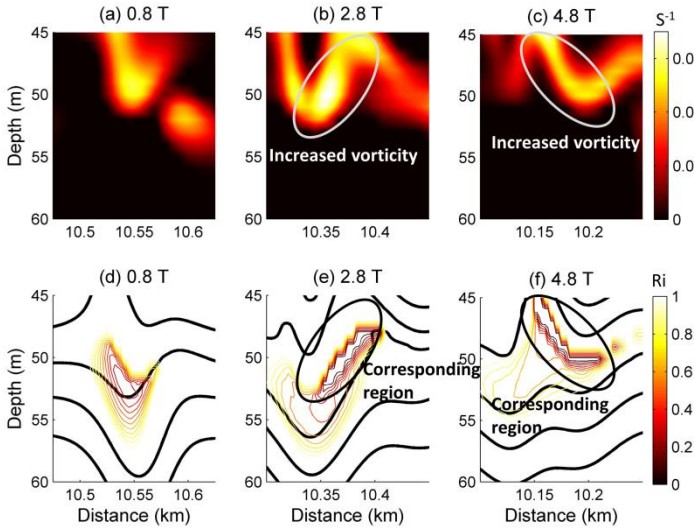

**Figure 17.** The vorticity spatial distributions for the case O5 (Δ = 0) at (a) 0.8 T, (b) 2.8 T, (c) and 4.8 T, and the

corresponding Ri values (ranges from 0 to 1)at (d) 0.8 T, (e) 2.8 T, and (f) 4.8 T.

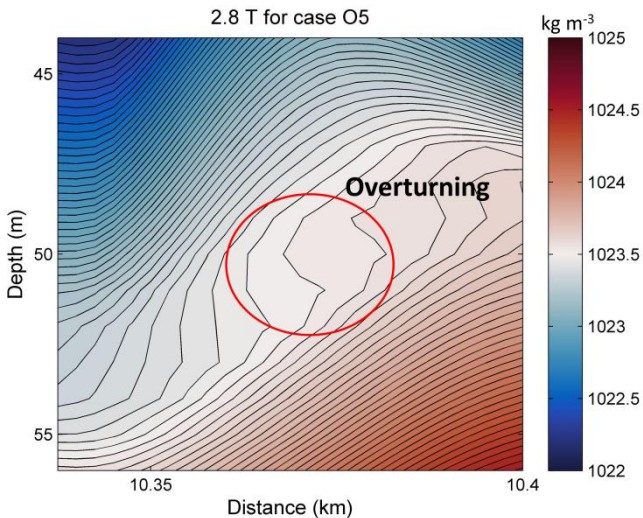

**Figure 18.** The density contour plot at 2.8 T for the case O5.

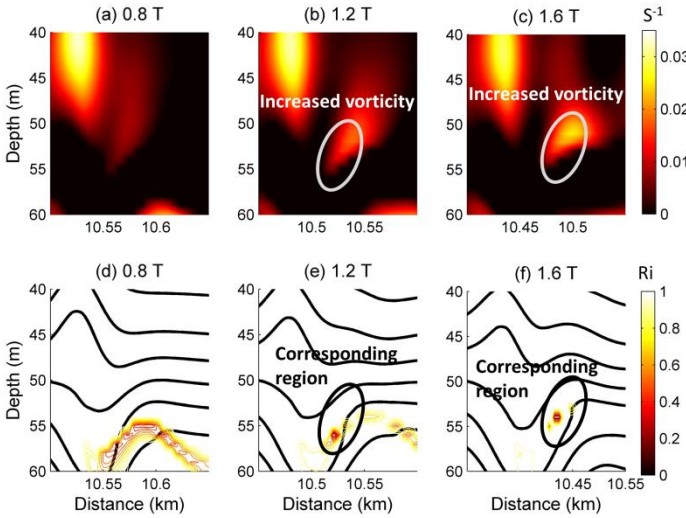

**Figure 19.** The vorticity spatial distributions for the case O9 ($\Delta = 2$) at 0.8 T (a), 1.2 T (b), and 1.6 T (c), and the corresponding Ri values (ranges from 0 to 1) at 0.8 T (d), 1.2 T (e), and 1.6 T (f).

As for the long-term behavior, the mode-2 ISW could adjust itself to adapt to new background conditions and experience a dramatic transformation with disintegration into a wave train (Grimshaw et al., 2010, Yuan et al., 2018). In our simulation, the mode-2 ISW was observed to adjust itself to the new background condition with a shear current. The high energy loss rate is in agreement with the observation by Shroyer et al. (2010). However, the mode-2 ISW might not be able to survive for long

time in situ because the background conditions in real ocean could be more complex and vary with time, leading to a background condition where a stable solution of mode-2 ISW does not exist.

**6 Conclusions**

We have presented the evolution process of mode-2 ISWs modulated by varied background shear currents with the MITgcm in this study. It was illustrated that the adjustment of the mode-2 ISWs in the

presence of background shear current occurs through the generation of forward-propagating long waves, amplitude-modulated wave packet, and an oscillating tail.

For comparison with the observation, a control experiment was conducted (case O5). ~36% of the total energy of the mode-2 ISW was lost at an average rate of 9 W m$^{-1}$, and this rate was in agreement with the observation of Shroyer et al. (2010). The mode-2 ISWs are highly dissipated in the presence of

shear currents, and it was consistent with the hypothesis given by Shroyer et al. (2010).In addition, five sets of experiments were introduced to assess the sensitivity of the evolution process to different

properties of the background shear currents in order to get general conclusion. We found that the polarity and direction of the background shear current have minor effect on the evolution of the mode-2 ISW. The amplitudes of the forward-propagating long wave and amplitude-modulated wave packet as well as the decaying of mode-2 ISWs' energy are proportional to the magnitude of shear but inversely proportional to the thickness of the shear layer. We also found that the oscillating tail and amplitude-modulated wave packet show symmetric variation trend in both offset upward and downward conditions, while the forward-propagating long wave was insensitive to the background shear current, and the shear layers centered at the mid-depth of pycnocline had much more pronounced energy loss of the mode-2 ISW compared to those cases where the shear layer centered away from the mid-depth of the pycnocline.

In future work, a possible avenue is the evolution of mode-2 ISW in time-varied background shear current, and the wave-mean flow interaction in this complicated flow fields, revealing the energy exchange process between waves and mean-flow. The other possible direction is the investigation of combination effect which is more close to field observations on the evolution of the mode-2 ISW, including the effect of background shear current, varying topography and shoaling pycnocline.

### Acknowledgments

Funding for this study was provided by National Key Research and Development Program of China (No. 2017YFA0604102), the Scientific and Technological Innovation Project Financially Supported by QNLM (No. 2016ASKJ12), National Key Research and Development Program of China (No. 2016YFC1401404), the National Natural Science Foundation of China (41528601, 41676006, 41421005, 41576189), Youth Innovation Promotion Association CAS, CAS Interdisciplinary Innovation Team "Oceanic Mesoscale Processes and Ecology Effects", Key Research Program of Frontier Science, CAS (QYZDB-SSW-DQC024), and the Strategic Pioneering Research Program of CAS (XDA11020104, XDA11020101). This study was supported by the High Performance Computing Center at the IOCAS. Constructive and helpful comments from the editor, Prof. Tatiana Talipova, and two anonymous reviews are gratefully acknowledged. The MITgcm program can be downloaded from the websites at http://mitgcm.org. The EOF codes can be downloaded from http://cn.mathworks.com/matlabcentral/fileexchange. The simulation data deposit for this paper needs high-capacity disk and is available on the request to Dr. Zhenhua

Xu by email.

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
