# Peer review of "The evolution of mode-2 internal solitary waves modulated by background shear currents"

_Nonlinear Processes in Geophysics, 2017_

## Referee Comment (RC1) · Anonymous Referee #1 · 7 Feb 2018

**1   General comments**

The article presents novel numerical results describing the adjustment of a mode-2 ISW due to a background shear flow. The authors vary the center of the background shear and measure the amount of energy lost from the ISW. The authors attempt to demonstrate that this energy is radiated into three different types of waves: a leading mode-1 wave, an oscillating tail, and an amplitude-modulated wave packet. Comparison to the work of Shroyer et al. (2010) is made throughout.

The paper is well structured and contains new results that are applicable to a wide audience. However, substantial work is required to bring the paper up to an international standard. For example, there is missing literature review on mode-1 ISWs in background currents and wave-mean flow interaction, various quantities are not carefully defined, and five cases do not suggest a fully developed study. Detailed suggestions and questions are below. I would like to see the article published, but the authors must address the following concerns.

**2 Major comments**

1. There are no references to similar studies of mode-1 ISWs in shear flow. Comparisons to lower mode internal waves should be made to better position this article within the literature. Suggestions include: Stastna and Lamb (2002), and Lamb (2010).

2. Other than Shroyer et al. (2010), what other references exist for mode-2 ISWs in background currents? Are there none? That seems surprising.

3. What predictions does theory make? Do the author's results match those of weakly nonlinear theory? It appears that KdV theory will apply and much can be learned by using standard techniques of KdV theory.

4. Do five cases provide sufficient information to make generalized claims about ISWs in shear flow? For comparison, Maderich et al. (2017) ran 35 cases of collisions of internal waves.

5. How were the values of the shear chosen? Were they chosen to match flow on the New Jersey shelf? What would happen if the shear was varied in terms of magnitude, or was oriented against the ISW (such that $U_1$ and/or $U_2$ were positive). What about shifting the shear upwards rather than down? What about $h_s$ compared to $h$? Please add some of these cases into the article. Further motivation for the values is required.
6. How was the background shear introduced? Was it continuously increased over a short duration of time or instantaneously turned on? What was numerically done to add the shear? The first paragraph of section 2.3 does not make it clear that the ISW was generated without the presence of the background shear. Please correct this.

7. Page 7, line 4. How was the amplitude of the wave calculated? Much of section 3 discusses the change in amplitude or compares different amplitudes, but it is unclear where this came from. $L$ is also not defined in the text.

8. What is the difference between the oscillating tail and the amplitude-modulated wave packet? To me they look identical. What are their defining features? How did you distinguish between them?

9. Page 9, lines 9-16. How does the complexity of the vorticity field imply higher energy transfer? How do you know that the energy in the radiating waves does not arise from the background mean flow? Have the authors read the literature on wave-mean flow interaction? This seems highly pertinent and must be included in the article. Lastly, how does a larger amplitude imply a higher energy? Are the authors assuming linearity? Is this applicable here? The use of the term applied (or implying) in these sentences is not justified, and begs further quantification. (see also lines 18-21 on page 11).

10. In connection to the previous comment, section 3.4 makes many of the same assumptions about how energy and wave amplitude are related. But this relation is unclear and non-trivial.

11. Page 9, lines 20-22. The consequent in the following conditional sentence does not following from the antecedent. "Based on the vorticity field shown in Fig. 6 (d), the generation of the oscillating tail and the forward-propagating long wave was continuously sustained by the energy of the ISW". A single snapshot does

not indicate anything about the continuous evolution of the radiating waves. I suggest adding a Hovmoller (space-time) plot to show the time varying nature of the waves. However, this is only possible if sufficient time resolution is available.

12. Page 13. How are $x_l$ and $x_r$ defined? The authors say they are the boundaries of the integration region, but don't specify where they come from. They are critical to the discussion of the size of the wave, and the choice of definition will have a large impact in the energy values.

13. Page 20, lines 9-11. What waves are transient to the introduction of the shear, and what are persistent? Much of this article seems to consist of the initial adjustment when the background shear is introduced. What is the long time behaviour of the mode-2 wave in the presence of shear? On another note, what can be said about the generation of a mode-2 ISW while shear is present?

**3 Specific comments**

1. Page 3, lines 12-14. Are there references?

2. Page 4, what evidence do you have that your solution is numerically convergent or accurate? Did you conduct grid refinement strategies? How were the viscosity parameters chosen? What motivation do you have for them?

3. Page 4, line 19. Please write the equation for $\Delta$ out explicitly.

4. Figure 1. Please define the cases in the main body of the article and not in the figure caption.

5. $h_{mix}$ and $l_{mix}$ are not defined in the text. Just in the caption for figure 3.

6. Is a lock-release of this form applicable on a field scale?

[Figure]

7. The colormap used in figures 4, 6, 8 is not good. Please change to something symmetric about a reference value. I suggest colormap 'balance' from Thyng et al. 2016. A matlab package is available for download.

8. What is being plotted in figure 4a? Is it temperature? Please make clear.

9. Text in figures 5, 6, etc. is too small and the resolution needs to be increased.

10. Put colorbars on all vorticity plots.

11. Page 10, lines 2-3. I do not see how the following sentence arises from the statement just prior to it, "Thus, the energy loss of the ISW caused by forward-propagating long waves occurs earlier than oscillating tail." I don't see how the forward-propagating wave happens earlier.

12. Figure 8. What is BLIW? Define this somewhere.

13. Page 11, line 16. Do the authors have any physical reason why "the forward-propagating long wave may not be affected by $\Delta$"?

14. Page 13, the vertical integrals can be written as $\int_{-H}^{0}$ rather than $\int_{-H(x)}^{0}$ since the bottom topography is flat.

15. Page 13, at what time did the initial mode-2 ISW contain 146.2 KJ m$^{-1}$? Just before the introduction of the shear?

16. Page 13, Define EOF and give a brief overview of its applicability here. Why is it applicable while normal mode decomposition is not?

17. Can you explain the periodicity of figure 10 b? Why is it a function of distance and not time? What about the long time behaviour (when the we wave approaches $x = 0$)?

18. What are the breakdowns of the energy flux in terms of $KE_f$, $APE_f$, and $W$? What happens before $t = 6$T?

19. Page 15, lines 11-13. I'm still not convinced that the following is true: "the high radiating flux before 12 T indicates the generation process of the amplitude-modulated wave packet and that the relative low radiating energy flux above 12 T is caused by the generation of an oscillating tail." The authors have yet to clearly show that the amplitude-modulated wave packet is short lived while the oscillating tail is persistent.

20. Page 16, line 13. It appears that the authors haven't been rigorous enough about how the energy is transferred. Why use the word 'suggests'?

21. Table 2. Should the units be KW m$^{-1}$?

22. Page 17, line 8. which observations where compared? and where? Were there more than the Shroyer et al. (2010) paper?

23. Figure 15. What field is being plotted? Density? Which isopycnal is being plotted?

24. The text at times is missing articles (such as 'the') and the tense is sometimes mixed up. Please review for grammar.

---

## Referee Comment (RC2) · Anonymous Referee #2 · 9 Feb 2018

**The evolution of mode-2 internal solitary waves modulated by background shear currents by Zhang *et al.**

The effect of a background shear current on the evolution of a mode-2 internal solitary wave is investigated using the MITgcm numerical model. Three features were identified due to the modulation of the mode-2 wave by the background shear, namely, (i) forward-propagating long waves, (ii) an amplitude modulated wave packet behind the mode-2 wave and (iii) an oscillating tail. The distance between the centre of the shear layer and the centre of the pycnocline was varied such that the distance went in incremental values from zero (no offset) to offsets in which the centre of the shear layer was below that of the pycnocline. It was shown that the forward-propagating waves were insensitive to the offset distance while the oscillating tail and the wave packet decreased in their respective amplitudes as the offset was increased. Implications for energy transfer and energy depletion of the original mode-2 wave are discussed and comparison to a related field study (Shroyer *et al.* 2010) are given.

The paper is original and makes some interesting findings, as such I am in favour of publication but unfortunately the paper is not suitable in it's present form. The following comments and suggestions are provided should the authors wish to rework the paper.

- 1) The paper is littered with grammatical and typographical errors. A thorough check is required.
- 2) Abstract lines 13-16 : this is not at all clear to the reader. The reader only knows what these features are AFTER reading the paper.
- 3) Abstract: The definition of delta is not clear e.g. which distance (shear or pycnocline centre) is divided by which ?
- 4) Abstract: long waves are said to be "robust" to delta. What does this mean ? Insensitive ? Not affected by ?
- 5) Introduction: Mode-2 waves have also been remotely observed please see and reference JACKSON, CHRISTOPHER R., et al. "Nonlinear Internal Waves in Synthetic Aperture Radar Imagery." *Oceanography*, vol. 26, no. 2, 2013, pp. 68–79. *JSTOR*, JSTOR, www.jstor.org/stable/24862037.
- 6) Introduction line 6: "in slope" not sure why the authors make specific reference to a slope here, e.g. can we infer that convex and concave are observed as much as one another in areas where there is not a slope ?
- 7) P4 line 4 define viscosities, what do the sub scripts stand for ?
- 8) P4 line 19 it would be useful to have a figure here explaining exactly what delta is. The authors may also like to consider adopting a similar definition and symbols to what others already use in the literature. For example see Neil Balmforth's work on identifying unstable modes in stratified flows.
- 9) Figure 1: The authors have chosen to set the centre of the pycnocline at mid depth but in the field this is not the case and others (e.g Olsthoorn et al 2013 and Carr et al 2015) have shown that the location of the pycnocline relative to mid depth has a crucial influence on the shape and form of a mode-2 wave. This warrants discussion.
- 10) Figure 2: The figure shows that the larger delta is, the smaller Ri can be. This is interesting. Can the authors explain this finding ? Has it been reported elsewhere ? Eg Balmforth again.
- 11) P 6 line 5. It is misleading to reference mode 1 work here as the initial condition (set up behind the gate) is different and in fact it is the initial condition that is crucial in generating a mode-2 wave (as opposed to mode-1). It would be more appropriate to reference just Brandt & Shipley along with mode-2 papers such as Olsthoorn et al 2013 and/or Deepwell & Statsna 2016, and/or Statsna et al 2015.

- 12) Figure 3: The authors have chosen to offset the shear centre downward of the pycnocline. Do they expect to see similar results (but symmetrically reversed) if it were to be offset in the upward direction ? Presumably as the pycnocline centre is at mid -depth. What would happen however if the pycnocline centre were not at mid depth ? Also the authors have chosen the shear such that the current in the top layer is in the same direction as the wave - this is similar to the overtaking cases in the work by Stastna et al 2015 and some comparison with that work should be given. Do the authors expect to see the same or different dynamics if the polarity of the shear current is reversed ?
- 13) P. 7 line 6 it'd be useful if cp were given and/or c presented in non dimensional form.
- 14) Figure 4 caption: (a) "wave form" is this temperature ? What quantity and scale is the colour bar ?
- 15) Page 9 text and figures it is difficult to see the forward propagating waves can this be improved ?
- 16) Page 13 line 8 what are x\_r and x\_l taken to be though ?
- 17) Page 14 line 19 confusing grammar suggests mode-1 are also short lived
- 18) Page 17 line 22 are the authors referring to the field here or their simulations ?
- 19) Figs 13 and 14 and related discussion. If shear instability is present would you not expect to see overturning isopycnals ?
- 20) Page 19 Line 6 onward. Nice discussion which makes things a lot clearer for the reader, may be this should be given much earlier in the paper.
- 21) Page 20 line 2. This is not clear there was no background shear in the papers cited in line 1. What do the authors mean here by shear ?

---

## Referee Comment (RC3) · T. Talipova (Referee) · 11 Feb 2018

Review on "The evolution of mode-2 internal solitary waves modulated by background shear currents" by Peiwen Zhang, Zhenhua Xu, Qun Li, Baoshu Yin, Yijun Hou, and Antony K. Liu

The evolution of the mode-2 internal solitary waves in fluid with stratification in density and shear flow is studied numerically with use MITgcm software. It is demonstrated that initial solitary wave disturbance splits on several wave groups: forwarding-propagated long waves, amplitude-modulated wave packet (which is identified as breather) and oscillating tail behind of soliton. Such groups attenuate with distance due to transfer processes, viscosity and shear instability. Interesting moment of this study is also the

difference in the location of pycnocline and shear flow. Obtained results are important for understanding processes related with mode-2 internal waves and I may recommend publishing given paper. Some comments: 1. In fact, the difference between oscillating tail and amplitude-modulated packet (breather) is not clear visible. For instance, the modal structure is discussed on Fig. 10, but it will be more useful to see modal structure on waves for different times. The first mode contributes in energy mainly, but what is a difference in amplitudes of modes in the pycnocline? 2. What is an origin of the forward-propagating long waves? Are they generated in the initial time only? 3. Usually oscillatory tail is generated behind solitary wave due to dissipation, and its energy is increased when soliton energy is decreased. Is it observed in numerical simulations?

Technical comments: i) Axis z is directed up. Looking on formulas (1) and (2) $z = 0$ corresponds to the fluid surface, but on Fig. 3 – to the fluid bottom. ii) Fig. 6. There is no vorticity scale.

I recommend publishing after minor revision.

---

## Author Comment (AC1) · 24 Apr 2018

**Response to Reviewer 1:**

**General comments**

The article presents novel numerical results describing the adjustment of a mode-2 ISW due to a background shear flow. The authors vary the center of the background shear and measure the amount of energy lost from the ISW. The authors attempt to demonstrate that this energy is radiated into three different types of waves: a leading mode-1 wave, an oscillating tail, and an amplitude-modulated wave packet. Comparison to the work of Shroyer et al. (2010) is made throughout.

The paper is well structured and contains new results that are applicable to a wide audience. However, substantial work is required to bring the paper up to an international standard. For example, there is missing literature review on mode-1 ISWs in back- ground currents and wave-mean flow interaction, various quantities are not carefully defined, and five cases do not suggest a fully developed study. Detailed suggestions and questions are below. I would like to see the article published, but the authors must address the following concerns.

**Response:**

Thank you very much for your constructive and helpful comments, which were extremely helpful in improving the manuscript. We have carefully read the comments and made substantial revisions accordingly as detailed in the following responses to specific questions. The NPG Language Editing service was contracted to review and polish the revised manuscript before submission. The main revisions are highlighted in the manuscript.

**Question 1**

There are no references to similar studies of mode-1 ISWs in shear flow. Comparisons to lower mode internal waves should be made to better position this article within the literature. Suggestions include: Stastna and Lamb (2002), and Lamb (2010).

**Response:**

In response to the reviewer's helpful comments, references related to mode-1 ISWs in shear flow have been added and discussed in the revision. Particularly, we have used the same method as that in *Lamb* (2010) to calculate the ISW energy in the presence of the background current. Some analysis methods in *Stastna and Lamb* (2002) are also used in the present study to investigate the variation of mode-2 wave properties in background shear current.

The related descriptions were added to the revision as follows (see also Page 2, Line 27 – Page 3, Line 4 in the main text):

"*The evolution of mode-1 ISWs in background shear current was extensively studied (Stastna and Lamb, 2002, Lamb, 2010, Grimshaw et al., 2005, Fructus et al., 2009). Lamb (2010) investigated the energetics of mode-1 ISWs in a background shear current, providing some methods commonly used to calculate the energy under that circumstance. Stastna and Lamb (2002) considered the effects of background current on mode-1 ISWs and discussed the properties of ISWs during the breaking process. In comparison, few works on mode-2 ISW in shear flow have been produced.*"

**Question 2**

Other than Shroyer et al. (2010), what other references exist for mode-2 ISWs in background currents? Are there none? That seems surprising.

**Response:**

Accordingly, the studies of mode-2 ISWs in the background current have been reviewed and summarized. Because of the difficulty to capture both the mode-2 waves and low-frequency background flows, to our knowledge, *Shroyer* et al. (2010)'s study is the sole one with the observational evidence based on which we can numerically examine the integrated evolution process of mode-2 ISWs in the background shear current.

The related descriptions were added to the revision as follows (see also Page 3, Lines

6 – 9 ; Page 3, Lines 19 – 21 in the main text):

"*Vlasenko* et al. *(2010) observed mode-2 ISW followed by mode-1 ISW in tidal flow in the South China Sea. Liu* et al. *(2013) investigated the generation and evolution of mode-2 ISW in the South China Sea and concluded that the mode-2 ISW might not propagate, evolve and persist for long time on the shelf.*"

"*Shroyer* et al. *(2010) is the sole one recording an integrated evolution process of mode-2 ISW in the background shear current, based on which we can numerically examine the modulation of mode-2 ISW in that circumstance.*"

**Question 3**

What predictions does theory make? Do the author's results match those of weakly nonlinear theory? It appears that KdV theory will apply and much can be learned by using standard techniques of KdV theory.

**Response:**

The initial properties of mode-2 ISW without the presence of a background shear current in our study is consistent with the prediction by KdV (Korteweg-de Vries) theory (Figure 1) (*Grimshaw* et al., 2007). As introduced by *Maderich* et al (2010), the weakly nonlinear KdV theory is suitable for slowly varying background conditions, but the intensive evolution process is not expected in the KdV framework, and as a result, a numerical modal approach is needed (*Grimshaw* et al., 2010; *Terletkska* et al., 2016; *Yuan* et al., 2018). The recent work by *Yuan* et al., (2018) observed the existence of a long mode-1 wave ahead of the mode-2 ISW during the evolution of mode-2 ISW over variable topography and found that this process cannot be characterized by KdV theory. Therefore, in the present study, we use the full-nonlinear and nonhydrostatic MITgcm model to examine the integrated evolution process of mode-2 ISW in the background shear current.

[Figure]

**Figure 1.** The characteristics of a single mode-2 ISW's (a) density field (1022 kg m$^{-3}$ – 1026 kg m$^{-3}$) and (b) vorticity (-0.035 s$^{-1}$ – 0.035 s$^{-1}$) in the absence of background shear current. The white lines in (a) demonstrate the theoretical solution of mode-2 ISW in KdV framework.

The related descriptions were added to the revision as follows (see also Page 2, Lines 19 – 23 in the main text):

"*Yuan* et al. *(2018) observed the existence of a long mode-1 wave ahead of mode-2 ISW during the evolution of mode-2 ISW over variable topography and found that this process cannot be characterized by KdV theory. The authors suggested using the MITgcm model, which can solve all modes to investigate the integrated evolution process of mode-2 ISWs in variable background conditions.*"

**Question 4**

Do five cases provide sufficient information to make generalized claims about ISWs in shear flow? For comparison, Maderich et al. (2017) ran 35 cases of collisions of internal waves.

**Response:**

Thank you for the constructive suggestion. Accordingly, several important factors were investigated to generalize our research in the literature, including the magnitude, thickness of shear layer, direction and symmetric offset of background shear current. The detailed configuration of the numerical simulation is given in Table 1.

**Table 1.** Summary of parameters of variable background shear currents. The depth and thickness of the pycnocline are denoted by $z_0$ and $h$. The thickness and depth of the shear layer are denoted by $h_s$ and $D_s$. The magnitude of the shear current is denoted by $U_d$. The *Por* indicates the polarity of the background shear current, and "+" means the polarity-reversal shear current. The orientation of the background shear current is indicated by *Ori*, and "+" means an opposing shear current.

| Case | $z_0$ | h | $h_s$ | $D_s$ | $U_d$ | Por | Ori |
|------|-------|-----|-------|-------|-------|-----|-----|
| O1 | 50 | 10 | 3 | 40 | 0.22 | (-) | (-) |
| O2 | 50 | 10 | 3 | 42.5 | 0.22 | (-) | (-) |
| O3 | 50 | 10 | 3 | 45 | 0.22 | (-) | (-) |
| O4 | 50 | 10 | 3 | 47.5 | 0.22 | (-) | (-) |
| O5 | 50 | 10 | 3 | 50 | 0.22 | (-) | (-) |
| O6 | 50 | 10 | 3 | 52.5 | 0.22 | (-) | (-) |
| O7 | 50 | 10 | 3 | 55 | 0.22 | (-) | (-) |
| O8 | 50 | 10 | 3 | 57.5 | 0.22 | (-) | (-) |
| O9 | 50 | 10 | 3 | 60 | 0.22 | (-) | (-) |
| H1 | 50 | 10 | 1.5 | 50 | 0.22 | (-) | (-) |
| H2 (O5) | 50 | 10 | 3 | 50 | 0.22 | (-) | (-) |
| H3 | 50 | 10 | 4.5 | 50 | 0.22 | (-) | (-) |
| H4 | 50 | 10 | 6 | 50 | 0.22 | (-) | (-) |
| H5 | 50 | 10 | 7.5 | 50 | 0.22 | (-) | (-) |
| U1 | 50 | 10 | 3 | 50 | 0.11 | (-) | (-) |
| U2 (O5) | 50 | 10 | 3 | 50 | 0.22 | (-) | (-) |
| U3 | 50 | 10 | 3 | 50 | 0.33 | (-) | (-) |
| U4 | 50 | 10 | 3 | 50 | 0.44 | (-) | (-) |

| | | | | | | | |
|---|---|---|---|---|---|---|---|
| **U5** | 50 | 10 | 3 | 50 | 0.55 | (-) | (-) |
| **D1** | 50 | 10 | 3 | 50 | 0.22 | (-) | (+) |
| **P1** | 50 | 10 | 3 | 50 | 0.22 | (+) | (-) |

The related descriptions were added to the revision as follows (see also Lines Page 3, Lines 26 – 30; Page 5, Lines 10 – 15 in the main text):

"*To reveal the sensitivity of the evolution of mode-2 ISWs to variable parameters of background shear currents, we introduced five sets of experiments (19 experiments in total) to generalize our research, including the magnitude, thickness of the shear layer, direction and symmetric offset of the background shear current.*"

"*In the sensitive experiments, the magnitude of the shear current is denoted by $U_d$ and defined as $|\Delta U|$. This value was varied from $0.5U_d$ to $2.5\ U_d$ in the sensitivity test. Similarly, the thickness of the shear layer was varied from $0.5\ h_s$ to $2.5\ h_s$. In cases D1 and P1, an opposing and polarity-reversal background shear currents were initialized for examination, respectively. We introduced an asymmetry parameter $\Delta$ to investigate the evolution of mode-2 ISWs in the offset background shear current (Carpenter et al., 2010).*"

**Question 5**

How were the values of the shear chosen? Were they chosen to match flow on the New Jersey shelf? What would happen if the shear was varied in terms of magnitude, or was oriented against the ISW (such that U1 and/or U2 were positive). What about shifting the shear upwards rather than down? What about hs compared to h? Please add some of these cases into the article. Further motivation for the values is required.

**Response:**

In the case O5, which was taken as a control experiment, the values of background shear current were chosen to match the observations in the New Jersey Shelf (*Shroyer*

et al., 2010). The evolution process and calculated energy loss rate for the control experiment were in relative agreement with the field observations.

According to the reviewer's suggestions in questions 4 and 5, we have run more sensitive experiments in the revision (14 additional experiments in total). The main results include the following:

**Orientation**

The background shear current oriented against the mode-2 ISW only slightly affects the amplitude of the forward-propagating long wave and oscillating tail but significantly influences the amplitude of the amplitude-modulated wave packet. The opposing background shear current slightly increased the energy loss of mode-2 ISW during the modulation.

**Magnitude of the shear**

The amplitudes of the forward-propagating long wave and amplitude-modulated wave packet are positively proportional to the magnitude of the background shear current, but the oscillating tail is not very sensitive to the increasing magnitude of the background shear current (Figure 2 (a)). The energy losses of mode-2 ISW are also positively proportional to the magnitude of the background shear current (Figure 3 (a)).

**Thickness of the shear layer**

The amplitudes of all three shear-induced wave structures are negatively proportional to the thickness of the shear layer (Figure 2 (b)). The energy losses of mode-2 ISW are also negatively proportional to the thickness of the shear layer (Figure 3 (b)).

**Offset upward**

When the shear current is offset upward, the variation of the oscillating tail and amplitude-modulated wave packet in amplitude show similar trends to the downward offset cases (Figure 2 (c) and (d)), but the forward-propagating long wave is still insensitive to the offset direction of the shear current (Figure 2 (c) and (d)). The energy losses of mode-2 ISW are also negatively proportional to the asymmetry parameter $\Delta$ in both upward and downward conditions (Figure 3(c) and (d)).

[revised manuscript text omitted]

**Question 6**

How was the background shear introduced? Was it continuously increased over a short duration of time or instantaneously turned on? What was numerically done to add the shear? The first paragraph of section 2.3 does not make it clear that the ISW

was generated without the presence of the background shear. Please correct this.

**Response:**

The initialization of the mode-2 ISW in section 2.3 has been refined accordingly. The background shear current was instantaneously turned on after the appearance of a stable mode-2 ISW in the absence of a background shear current. Then, the velocity field of the background shear current was superimposed on the model.

The related descriptions were added to the revision as follows (see also Page 7, Lines 3 – 4, Page 7 Lines 14 – 15 in the main text):

"*A rank-ordered mode-2 ISW train was generated by the "lock-release" method without background current.*"

"*At 6000 s after the initialization of numerical model, the velocity field of the background shear current was superimposed on the model.*"

**Question 7**

Page 7, line 4. How was the amplitude of the wave calculated? Much of section 3 discusses the change in amplitude or compares different amplitudes, but it is unclear where this came from. L is also not defined in the text.

**Response:**

The definitions and calculations of the amplitude have been clarified accordingly, and the definition of *L* has been added in the manuscript.

The related descriptions were added to the revision as follows (see also Page 7, Lines 10 – 14 in the main text):

"*The amplitudes of shear-induced waves were defined as maximum isopycnal displacement (Stastna and Lamb, 2002). The amplitude A of mode-2 ISWs was defined as the maximum displacement of the upper and lower isopycnals, which are equal in*

*the initial state (Terletska et al., 2016). The wavelength L was defined as the width of the wave at half of the amplitude of the mode-2 ISW in the initial state.*"

**Question 8**

What is the difference between the oscillating tail and the amplitude-modulated wave packet? To me they look identical. What are their defining features? How did you distinguish between them?

**Response:**

Detailed definitions of the oscillating tail and amplitude-modulated wave packet have been added in the revision. The related figures have been improved to clearly demonstrate their differences. The amplitude-modulated wave packet appeared at the end of oscillating tail as steady-state envelopes *(Terletska* et al., 2016), and it could be clearly observed at the end of the oscillating tail (Figure 4). The amplitude-modulated wave packet (Figure 5) was defined as a pulsating wave packet propagated with a steady-state envelope, inside which the waves oscillate freely with different amplitudes (*Terletska* et al., 2016).

[Figure]

**Figure 4**: The evolution process of mode-2 ISW in case O5 at 14 T, where the 'ot' and 'am' denoted the oscillating tail and amplitude-modulated wave packet, respectively (see also FIigure 4(d) in the manuscript).

[Figure]

**Figure 5.** The envelopes of the amplitude-modulated wave packet in the case O5 ($\Delta = 0$) at (a) 12 T, (b) 20 T and

(c) 28 T, where the mean density isopycnal of upper and lower layer (green line) are plotted.

The related descriptions were added to the revision as follows (see also Page 10, Lines 6 – 11, Page 10, Lines 19 – 21 in the main text):

"*It also caused the asymmetrical distribution of the vorticity in the horizontal, which is associated with the generation of forward-propagating long waves and the amplitude-modulated wave packet (Fig. 4 (b)), and the latter was defined as a pulsating wave packet (Clarke et al., 2000). The pulsating wave packet propagated with a steady-state envelope, inside which the waves oscillate freely with different amplitudes (Terletska et al., 2016).*"

"*The oscillating tail caused by the shear was visible between the mode-2 ISW and the amplitude-modulated wave packet (Fig. 5 (d)) at 14 T after the initialization of the background shear current, and it was a radiated mode-1 oscillatory disturbance trailing mode-2 ISW (Stamp and Jacka, 1995).*"

**Question 9**

Page 9, lines 9-16. How does the complexity of the vorticity field imply higher energy transfer? How do you know that the energy in the radiating waves does not arise from the background mean flow? Have the authors read the literature on wave-mean flow interaction? This seems highly pertinent and must be included in the article. Lastly, how does larger amplitude imply a higher energy? Are the authors assuming linearity? Is this applicable here? The use of the term applied (or implying) in these sentences is not justified, and begs further quantification. (See also lines 18-21 on page 11).

**Response:**

Thank you for your suggestion. The vague description of the relationship between the complexity of vorticity and the energy transfer has been revised. The linear relation between energy and amplitude is inappropriate here, and it has been improved accordingly.

We reviewed some classic works on wave-mean flow interaction. As introduced by *Grimshaw* (1984), the energy of radiating waves is generally exchanged with the mean flow. Therefore, we compared the energy loss of mode-2 ISWs with the total energy of radiating waves and found they were nearly the same in quantity. This finding means that the energy from the background mean flow makes a relatively small contribution to the radiating waves. To reveal the energy transfer process in wave-mean flow interactions require a detailed analysis, but this is beyond the main scope of present study, which focused on the evolution of mode-2 ISWs in the background shear current. A comment has been added to the manuscript that notes the importance of wave-flow interaction investigations in the near future.

The related descriptions were added to the revision as follows (see also Page 10, Lines 11 – 18; Page 15, Lines 6 – 8; Page 27, Lines 21 – 26 in the main text):

"*To the aft of the ISW, the shear induced by the background currents lead to the deformation of the vortex dipole, and an increasing complexity of the vorticity field implied an intensive adjustment occurred. As illustrated in Fig. 5 (c), 10 T after the initialization of the background shear current, the vorticity of the mode-2 ISW is*

*redistributed to adapt to the background condition. In this process, which is related to the generation of an amplitude-modulated wave packet and a forward-propagating long wave, the vortex of the ISW shrank. The forward-propagating long wave and amplitude-modulated wave packet can be seen in Fig. 5 (c) with amplitudes of 0.25 m and 1.8 m respectively, and the latter was clearly observed at the rear of the mode-2 ISW.*"

"*The small amplitude of the oscillating tail and amplitude-modulated wave packet indicate that the modulation could be weaker when $\Delta$ increased, and the weakening of the oscillating tail makes the amplitude-modulated wave packet more visible.*"

"*The presence of background shear current generally caused the energy exchange between waves and mean flow (Grimshaw, 1984). We compared the energy loss of mode-2 ISW with the total energy of radiating wave and found that they were nearly the same in quantity. It means the energy from the background mean flow makes a relatively small contribution to the radiating waves. This avenue represents a possible direction for further investigation of the wave-mean flow interaction in complicated flow fields.*"

**Question 10**

In connection to the previous comment, section 3.4 makes many of the same assumptions about how energy and wave amplitude are related. But this relation is unclear and non-trivial.

**Response:**

The descriptions in section 3.4 have been improved accordingly. We have also optimized the structure of the manuscript to include the relationship between the amplitudes of shear-induced waves with different parameters of background shear currents, and the energy analysis has been moved to the following sections.

The related descriptions were added to the revision as follows (see also Page 15, Lines 17 – 26 in the main text):

"*The modulation caused by background shear currents was weakened as the Δ increased, corresponding to the decreased amplitude of the amplitude-modulated wave packet and oscillating tail. The amplitude-modulated wave packet has the highest amplitude among all cases compared to those of the other two wave forms. The amplitude of the amplitude-modulated wave packet decreased from 1.8 to 1 m monotonically, and the amplitudes of the oscillating tails decreased from 0.85 to 0.25 m between the case O5 (Δ = 0) and the case O7 (Δ = 1) but remained stable between the case O7 (Δ = 1) and case O9 (Δ = 2), indicating that the amplitude-modulated wave packet were more sensitive to the Δ than the oscillating tail. As expected, the ratio between the amplitude of modulated wave packet and oscillating tails increased from 2.1 in the case O5 to 4 in the case O9, so the amplitude-modulated wave packet became more distinct in case O9.*"

**Question 11**

Page 9, lines 20-22. The consequent in the following conditional sentence does not following from the antecedent. "Based on the vorticity field shown in Fig. 6 (d), the generation of the oscillating tail and the forward-propagating long wave was continuously sustained by the energy of the ISW". A single snapshot does not indicate anything about the continuous evolution of the radiating waves. I suggest adding a Hovmoller (space-time) plot to show the time varying nature of the waves. However, this is only possible if sufficient time resolution is available.

**Response:**

Thank you for your suggestion. A Hovmöller plot has been made to illustrate the time-varying nature of the evolution process of the mode-2 ISW (Figure 6). The oscillating tail and forward-propagating long wave could be found to evolve continuously. Related descriptions have been added to the manuscript.

[Figure]

**Figure 6**. Hovmöller plot of horizontal velocity without background shear current at the surface. The mode-2 ISW, forward-propagating long wave, oscillating tail and amplitude-modulated wave packet are denoted by 'm2', 'fp', 'ot' and 'am', respectively. The color bar ranges from -0.034 to 0.034 m/s.

The related descriptions were added to the revision as follows (see also Page 11, Lines 1 - 3; Page 11, Lines 5 – 8 in the main text):

"*A Hovmöller plot (Fig. 7) of horizontal velocity without the background shear current at the surface was plotted (Lamb et al., 2014). The forward-propagating long wave, oscillating tail and amplitude-modulated wave packet were found to propagate persistently.*"

"*Based on the vorticity field (Fig. 6 (d)) and the time-space varying nature (Fig. x), the generation of the oscillating tail and forward-propagating long wave was continuously sustained by the energy of the ISW. Therefore, they have the potential to drain the energy from an ISW over a long time scale.*"

**Question 12**

Page 13. How are $x_l$ and $x_r$ defined? The authors say they are the boundaries of the

integration region, but don't specify where they come from. They are critical to the discussion of the size of the wave, and the choice of definition will have a large impact in the energy values.

**Response:**

The definitions of $x_l$ and $x_r$ have been clarified. $x_l$ and $x_r$ are denoted the as the left and right boundaries, respectively, where the available potential energy flux equals zero (*Lamb*, 2010).

The related descriptions were added to the revision as follows (see also Page 17 Lines 6 - 8 in the main text):

"$x_r$ and $x_l$ are the boundary locations of the integration region, and x satisfies $x_l \leq x \leq x_r$. During the calculation of the wave energy, $x_r$ and $x_l$ are denoted as the left and right boundaries, respectively, where the available potential energy flux equals zero (Lamb, 2010)"

**Question 13**

Page 20, lines 9-11. What waves are transient to the introduction of the shear, and what are persistent? Much of this article seems to consist of the initial adjustment when the background shear is introduced. What is the long time behavior of the mode-2 wave in the presence of shear? On another note, what can be said about the generation of a mode-2 ISW while shear is present?

**Response:**

The amplitude-modulated wave packet, oscillating tail and forward-propagating long wave were observed to be persistent (Figure 6). The forward-propagating long wave and oscillating tail are generated persistently due to the continuous energy input from mode-2 ISW (Figure 6). The amplitude-modulated wave packet is generated transiently because it no longer receives energy from the mode-2 ISW after it propagates away.

As for the long-term behaviour, the mode-2 ISW could adjust itself to adapt to new background conditions if there exists a stable solution for the mode-2 ISW. In contrast, the energy of mode-2 ISW would radiate away if no stable solution for the mode-2 ISW is allowed in that circumstance. This dynamic also indicates that the mode-2 ISW could generate in an appropriate background shear current where a stable solution exists. A detailed discussion has been added in the manuscript.

The related descriptions were added to the revision as follows (see also Page 26 Lines 5 – 14 in the main text):

"*As for the long-term behaviour, the mode-2 ISW could adjust itself to adapt to new background conditions and experience a dramatic transformation with disintegration into a wave train (Grimshaw et al., 2010, Yuan et al., 2018). In our simulation, the mode-2 ISW was observed to adjust itself to the new background condition with a shear current. The high energy loss rate is in agreement with the observation by Shroyer et al. (2010). However, the mode-2 ISW might not be able to survive for long time in situ because the background conditions in real ocean could be more complex and vary with time, leading to a background condition where a stable solution of mode-2 ISW does not exist. It is also possible that the mode-2 ISW could generate in an appropriate background shear current if a stable solution exists in that condition.*"

**Specific comments**

**Question 1**

Page 3, lines 12-14. Are there references?

**Response:**

Related references have been cited.

**Question 2**

Page 4, what evidence do you have that your solution is numerically convergent or accurate? Did you conduct grid refinement strategies? How were the viscosity

parameters chosen? What motivation do you have for them?

**Response:**

The total energy of simulation domain has been integrated, and we found that it was convergent and stable. The initial mode-2 ISW in the absence of a background shear current was also in agreement with the KdV theory (Figure 1).

Second, no grid refinement strategies were applied. A high resolution was introduced to assure that the modulation of the mode-2 ISW in the background shear current could be clearly observed. The time-step of the simulation was set 0.4 s to satisfy the *Courant-Friedrichs-Lewy* condition.

The choice of viscosity was set following previous studies with similar field scales (*Grisouard et al.*, 2010, *Xie et al.*, 2015). The choice of viscosity aims to avoid breaking and ensure the model can run smoothly (*Guo* and *Chen*, 2012, *Yuan* et al., 2018).

**Question 3**

Page 4, line 19. Please write the equation for delta out explicitly.

**Response:**

The equation for Δ has been added.

The related descriptions were added to the revision as follows (see also Page 5, Lines 15 – 18 in the main text):

"*The asymmetry parameter Δ (Carpenter* et al.*, 2010) is defined as follows:*

$$\Delta = \frac{D_S - z_0}{h/2}$$

*where $D_S$ denotes the depth of shear centre and h denotes the thickness of pycnocline.*"

**Question 4**

Figure 1. Please define the cases in the main body of the article and not in the figure caption.

**Response:**

A detailed introduction of experiments has been added in the revision.

The related descriptions were added to the revision as follows (see also Lines Page 5, Lines 18 – 20 in the main text):

"*The Δ was varie*d *from -2 to 2 (case O1 to case O9) to investigate the evolution of the mode-2 ISW in the offset background shear current.*"

**Question 5**

$h_{mix}$ and $l_{mix}$ are not defined in the text. Just in the caption for figure 3.

**Response:**

The definitions have been added.

The related descriptions were added to the revision as follows (see also Page 7, Lines 5 – 7 in the main text):

"*Figure 3 demonstrates the configurations of the simulation domains. A mixed region was set to be symmetric around the centerline of the pycnocline at the right end, and its length $l_{mix}$ and height $h_{mix}$ were 375 m and 25 m, respectively.*"

**Question 6**

Is a lock-release of this form applicable on a field scale?

**Response:**

The mode-2 ISW generated by 'lock-release' method in our simulation remains stable and demonstrates a symmetric nature in agreement with the theory (Figure 1) and consistent with the observations (*Shoryer* et al., 2010).

**Question 7**

The colormap used in figures 4, 6, 8 is not good. Please change to something symmetric about a reference value. I suggest colormap 'balance' from Thyng et al. 2016. A matlab package is available for download.

**Response:**

We appreciate that constructive suggestion from the reviewer. We have applied the 'balance' colormap in the related figures accordingly.

**Question 8**

What is being plotted in figure 4a? Is it temperature? Please make clear.

**Response:**

Figure 4a shows the density field of mode-2 ISW. The caption has been modified.

**Question 9**

Text in figures 5, 6, etc. is too small and the resolution needs to be increased.

**Response:**

Revised accordingly.

**Question 10**

Put color bars on all vorticity plots.

**Response:**

Added.

**Question 11**

Page 10, lines 2-3. I do not see how the following sentence arises from the statement

just prior to it, "Thus, the energy loss of the ISW caused by forward-propagating long waves occurs earlier than oscillating tail." I don't see how the forward-propagating wave happens earlier.

**Response:**

This sentence has been improved in the revision. As shown in the Hovmöller plot (Figure 6), the forward-propagating long wave and amplitude-modulated wave packet generated simultaneously. The oscillating tail appeared after the amplitude-modulated wave packet propagated away from the mode-2 ISW, indicating that the energy loss caused by oscillating tail happened later than forward-propagating long wave.

The related descriptions were added to the revision as follows (see also Page 11, Lines 8 – 11 in the main text):

"*It should be noted that the forward-propagating long wave and amplitude-modulated wave packet generated simultaneously, while the oscillating tail appeared after the amplitude-modulated wave packet propagated away from the mode-2 ISW. Thus, the energy loss of the ISW caused by forward-propagating long waves occurs earlier than the oscillating tail.*"

**Question 12**

Figure 8. What is BLIW? Define this somewhere.

**Response:**

The word 'BLIW' was changed to 'amplitude-modulated wave packet', which is more easily understood.

**Question 13**

Page 11, line 16. Do the authors have any physical reason why "the forward-propagating long wave may not be affected by delta"?

**Response:**

As suggested by reviewer, we carried out sufficient experiments to further investigate the forward-propagating long wave that was generated by the collapse of mixing induced by shear instability. The results show that the amplitude of the forward-propagating long wave is proportional to the magnitude of the shear current, indicating that the forward-propagating long wave was affected by the strength of shear. Δ denoted the offset extent of background shear current, and the strength of shear remains unchanged when Δ varied. Therefore, the variation in Δ has no effect on the forward-propagating long wave.

The related descriptions were added to the revision as follows (see also Page 24, Lines 11 – 15 in the main text):

"*The results in section 3.8 show that the amplitude of the forward-propagating long wave is proportional to the magnitude of the shear current, indicating that the forward-propagating long wave was affected by the strength of shear. Δ denotes the offset extent of the background shear current, and the strength of the shear remains unchanged when Δ varied. Therefore, the forward-propagating long wave was insensitive to variation in Δ.*"

**Question 14**

Page 13, the vertical integrals can be written as $\int_{-H}^{0}$ rather than $\int_{-H(x)}^{0}$ since the bottom topography is flat.

**Response:**

Accepted.

**Question 15**

Page 13, at what time did the initial mode-2 ISW contain 146.2 KJm$^{-1}$? Just before the introduction of the shear?

**Response:**

Yes. The description of initial energy has been improved.

The related descriptions were added to the revision as follows (see also Page 17, Lines 10 – 11 in the main text):

"*The total energy of the initial mode-2 ISW just before the introduction of the shear calculated by the above expressions was 146.2 KJ m$^{-1}$*"

**Question 16**

Page 13, Define EOF and give a brief overview of its applicability here. Why is it applicable while normal mode decomposition is not?

**Response:**

A brief overview of EOF has been added. The normal mode decomposition is not suitable since the flow field in our simulations appears to be highly nonlinear in nature (*Venayagamoorthy* and *Fringer*, 2007).

The related descriptions were added to the revision as follows (see also Page 17 Line 23 – Page 18, Line 2 in the main text):

"*The EOF (empirical orthogonal function) method was applied to the modal decomposition. EOF is commonly used for mode decomposition and space-time-distributed datasets examination in oceanography (Venayagamoorthy and Fringer, 2007). The normal mode decomposition is not suitable for the analysis of forward-propagating long waves, amplitude-modulated wave packets or oscillating tails since the flow field in our simulations appears to be highly nonlinear in nature (Venayagamoorthy and Fringer, 2007)*"

**Question 17**

Can you explain the periodicity of figure 10 b? Why is it a function of distance and

not time? What about the long time behaviour (when the wave approaches x = 0)?

**Response:**

We revised Figure 10 to give the variation with a function of time (Figure 7). The periodicity was caused by the superimposition of the energy of the mode-1 oscillating tail and the trailing mode-2 energy. These features trailed the mode-2 ISW with different wavelengths and therefore exhibited periodicity. The long-term behaviour of wave structure can be viewed in the response to major question 13.

[Figure]

**Figure 7.** Percent contributions of mode-1 and mode-2 to the total kinetic energy in case ($\Delta = 0$) from 0 to 30 T at the (a) front and (b) rear of the mode-2 ISW.

**Question 18**

What are the breakdowns of the energy flux in terms of KEf , APEf , and W? What happens before t = 6T?

**Response:**

The breakdowns of the energy flux have been plotted accordingly (Figure 8), and we also refined Figure 11 in the previous manuscript to show the energy flux before 6 T.

The pressure perturbation energy flux dominates the energy flux both in the front and rear, and this phenomenon accords with *Lamb* and *Nguyen* (2009). Before 6 T, the mode-2 ISW deformed due to the shear effect, and the forward-propagating long wave and amplitude-modulated wave packet were not generated completely yet.

[Figure]

**Figure 8.** The breakdowns of the energy flux in terms of $W$, $KE_f$, and $APE_f$.

The related descriptions were added to the revision as follows (see also Page 20, Lines 4 – 7 in the main text):

"*The pressure perturbation generally make the largest instantaneous contributions to the total energy flux (Lamb and Nguyen, 2009; Venayagamoorthy and Fringer, 2005). For an ISW, the pressure perturbation term could be dominant (Lamb 2007). Since we focused on the energy loss of the mode-2 ISW, a total energy flux was analysed in the following paragraph.*"

**Question 19**

Page 15, lines 11-13. I'm still not convinced that the following is true: "the high radiating flux before 12 T indicates the generation process of the amplitude modulated

wave packet and that the relative low radiating energy flux above 12 T is caused by the generation of an oscillating tail." The authors have yet to clearly show that the amplitude-modulated wave packet is short lived while the oscillating tail is persistent.

**Response:**

We plotted the energy flux in the rear of the mode-2 ISW induced by amplitude-modulated wave packet and oscillating tail, respectively (Figure 9). The amplitude-modulated wave packet has a relative high energy flux during the early stage of modulation, and an oscillating tail with a small energy flux appeared after 12 T.

[Figure]

**Figure 9.** The total energy flux in the rear of the mode-2 ISW and the energy fluxes induced by the

amplitude-modulated wave packet and oscillating tail.

**Question 20**

Page 16, line 13. It appears that the authors haven't been rigorous enough about how the energy is transferred. Why use the word 'suggests'?

**Response:**

This sentence has been refined accordingly.

The related descriptions were revised as follows (see also Page 21, Lines 6 – 8 in the main text):

"*Combining the results of the energy flux in front and rear of the mode-2 ISW (Fig. 13), thus, in the early stage of modulation, the amplitude-modulated wave packet could make a larger contribution to the energy transfer process.*"

**Question 21**

Table 2. Should the units be KW m$^{-1}$?

**Response:**

The units of energy loss rates were W m$^{-1}$, and it was in consistent with the observations (10 W m$^{-1}$) of *Shoyer* et al., (2010)

**Question 22**

Page 17, line 8. which observations where compared? and where? Were there more than the Shroyer et al. (2010) paper?

**Response:**

The initial expression has been improved. The wavelength and amplitude of the mode-2 ISW were chosen to match the observations in the New Jersey Shelf (*Shroyer* et al., 2010).

The related descriptions were added to the revision as follows (see also Page 22, Lines 6 – 7 in the main text):

"*In the simulation of the present study, the wavelengths and amplitudes of the mode-2 ISWs were selected to be comparable to the observation of Shroyer* et al. *(2010) on the New Jersey Shelf.*"

**Question 23**

Figure 15. What field is being plotted? Density? Which isopycnal is being plotted?

**Response:**

The isopycnal of mean density was plotted, and the caption has been improved.

**Question 24**

The text at times is missing articles (such as 'the') and the tense is sometimes mixed up. Please review for grammar.

**Response:**

We have checked and improved the expressions of the manuscript. The NPG Language Editing service was contracted to review and polish the revised manuscript before the submission.

---

## Author Comment (AC2) · 24 Apr 2018

**Response to Reviewer 2:**

**General comment:**

The effect of a background shear current on the evolution of a mode-2 internal solitary wave is investigated using the MITgcm numerical model. Three features were identified due to the modulation of the mode-2 wave by the background shear, namely, (i) forward-propagating long waves, (ii) an amplitude modulated wave packet behind the mode-2 wave and (iii) an oscillating tail. The distance between the centre of the shear layer and the centre of the pycnocline was varied such that the distance went in incremental values from zero (no offset) to offsets in which the centre of the shear layer was below that of the pycnocline. It was shown that the forward-propagating waves were insensitive to the offset distance while the oscillating tail and the wave packet decreased in their respective amplitudes as the offset was increased. Implications for energy transfer and energy depletion of the original mode-2 wave are discussed and comparison to a related field study (Shroyer et al. 2010) is given.

The paper is original and makes some interesting findings, as such I am in favor of publication but unfortunately the paper is not suitable in its present form. The following comments and suggestions are provided should the authors wish to rework the paper.

**Response:**

Thank you for your encouraging and constructive comments, which greatly contributed to improving the manuscript. We carefully read and considered the comments and made substantial revisions. We hope you find these revisions acceptable, and we highly appreciate your suggestions and comments. We highlight the main revisions in the manuscript, and the important points are described below.

**Question 1**

The paper is littered with grammatical and typographical errors. A thorough check is required.

**Response:**

We have checked and improved the expressions of the revised manuscript. The NPG Language Editing service was contracted to review and polish the revised manuscript before submission.

**Question 2**

Abstract lines 13-16 : this is not at all clear to the reader. The reader only knows what these features are AFTER reading the paper.

**Response:**

Thanks for your suggestion. The abstract has been rewritten and improved accordingly.

The related descriptions were revised in the revision as follows (see also Page 1, Lines 12 – 26 in the main text):

"*The evolution of the mode-2 internal solitary waves (ISWs) modulated by background shear currents was investigated numerically. The sensitivity of modulation to the direction, polarity, magnitude, and shear layer thickness of the background shear current was assessed. In addition, the background shear currents were set to overlap or offset the pycnocline centre to investigate the effects on modulation. During the modulation, three observed shear-induced wave structures were categorized as the forward-propagating long wave, oscillating tail and amplitude-modulated wave packet. The amplitudes of the forward-propagating long wave and amplitude-modulated wave packet are proportional to the magnitude of shear but inversely proportional to the thickness of the shear layer. The oscillating tail and amplitude-modulated wave packet show symmetric variation when the background shear current is offset upward or downward, while the forward-propagating long wave was insensitive to the background shear current. The modulation is unaffected by the direction and polarity of shear. We compared the control experiment to the observations of Shroyer* et al. *(2010). In the first 30 periods, ~36% of the total energy was lost at an average rate of 9 W m$^{-1}$, consistent with the*

*results of Shroyer* et al. *(2010), who speculated that mode-2 ISWs are highly dissipated in the background shear current.*"

**Question 3**

Abstract: The definition of delta is not clear e.g. which distance (shear or pycnocline centre) is divided by which ?

**Response:**

The definition of Δ has been clarified.

The related descriptions were revised in the revision as follows (see also Page 1, Lines 14 – 16; Page 5, Lines 15 - 17 in the main text):

"*In addition, the background shear currents were set to overlap or offset the pycnocline centre to investigate the effects on modulation.*"

"*The asymmetry parameter Δ (Carpenter* et al*., 2010) is defined as follows:*

$$\Delta = \frac{D_S - z_0}{h/2}$$

*where $D_S$ denotes the depth of shear centre and h denotes the thickness of pycnocline.*"

**Question 4**

Abstract: long waves are said to be "robust" to delta. What does this mean? Insensitive ? Not affected by ?

**Response:**

'Insensitive' is more suitable, and the description has been changed accordingly.

The related descriptions were added to the revision as follows (see also Page 1, Lines 20 – 24 in the main text):

"*The oscillating tail and amplitude-modulated wave packet show symmetric variation when the background shear current is offset upward or downward, while the forward-propagating long wave was insensitive to the background shear current.*"

**Question 5**

Introduction: Mode-2 waves have also been remotely observed please see and reference JACKSON, CHRISTOPHER R., et al. "Nonlinear Internal Waves in Synthetic Aperture Radar Imagery." *Oceanography*, vol. 26, no. 2, 2013, pp. 68–79. *JSTOR*, JSTOR, www.jstor.org/stable/24862037.

**Response:**

Thank you for your suggestion. This reference has been included to provide evidence for the existence of mode-2 ISWs (see also Page 2, Line 5 in the revised manuscript).

**Question 6**

Introduction line 6: "in slope" not sure why the authors make specific reference to a slope here, e.g. can we infer that convex and concave are observed as much as one another in areas where there is not a slope ?

**Response:**

We revised this sentence following *Yang* et al. (2010) to clearly summarize the observation of mode-2 ISWs. As introduced by *Yang* et al. (2010), a concave slope is seldom observed because it requires a 'thick' middle layer, and this stratification is rare on the continental slope and shelf.

The related descriptions were added to the revision as follows (see also Page 2, Lines 9 – 10 in the main text):
"*In contrast, concave mode-2 ISWs are seldom observed because the stratification with a thick middle layer is rare (Yang* et al*., 2010)*"

**Question 7**

P4 line 4 - define viscosities, what do the sub scripts stand for ?

**Response:**

We added the definition of viscosities in the revision. The subscripts '*v*' and '*h*' stand for 'vertical' and 'horizontal', respectively.

The related descriptions were added to the revision as follows (see also Page 4,Lines 18 - 19 in the main text):

"*The viscosity parameters were set to* $10^{-3}$ $m^2 s^{-1}$ *for horizontal viscosity* $v_H$ *and* $10^{-4}$ $m^2 s^{-1}$ *for the vertical viscosity* $v_v$ *in the present study.*"

**Question 8**

P4 line 19 - it would be useful to have a figure here explaining exactly what $\Delta$ is. The authors may also like to consider adopting a similar definition and symbols to what others already use in the literature. For example see Neil Balmforth's work on identifying unstable modes in stratified flows.

**Response:**

We appreciate your constructive suggestion. We followed *Carpenter*, *Balmforth* and *Lawernce* (2010) to introduce the definition of an asymmetry parameter $\Delta$ to describe the asymmetry of the background shear current. We also improved Figure 3 in the revised manuscript to demonstrate the asymmetry parameter $\Delta$.

**Question 9**

Figure 1: The authors have chosen to set the centre of the pycnocline at mid depth but in the field this is not the case and others (e.g Olsthoorn et al 2013 and Carr et al 2015) have shown that the location of the pycnocline relative to mid depth has a crucial influence on the shape and form of a mode-2 wave. This warrants discussion.

**Response:**

As suggested by *Olsthoorn* et al. (2013), the essential patterns of the mode-2 ISW generation processes are the same for both asymmetric and symmetric conditions, suggesting that the basic structure of a mode-2 ISW with an offset pycnocline is similar to that for a mid-depth pycnocline. However, the asymmetric stratification can amplify the existing instability and induce asymmetrical instability (*Olsthoorn* et al., 2013), leading to more complicated circumstances, which makes it difficult to investigate the modulation process of mode-2 ISWs due to the presence of shear currents. To examine the influences of the background shear currents on the evolution process of a mode-2 ISW, a symmetric stratification was used in the present work following previous mode-2 works (*Terletska* et al., 2016; *Deepwell* and *Stastna*, 2016; *Deepwell* et al., 2017).

**Question 10**

Figure 2: The figure shows that the larger Δ is, the smaller Ri can be. This is interesting. Can the authors explain this finding ? Has it been reported elsewhere ? Eg Balmforth again.

**Response:**

As introduced by *Lamb* (2014), the Richardson number is defined as $Ri = N^2/u_z^2$, where $N^2$ is the buoyancy frequency and $u_z$ is the shear. The shear remains unaffected to the position of the shear centre. In larger Δ cases, the shear centre was offset from the pycnocline centre. A relatively small $N^2$ appears at this location, but the shear remains unchanged, causing a smaller Ri.

A similar result was observed by *Lamb* and *Farmer* (2011). In their work, a smaller *Ri* number could be found when the shear centre was located farther from the centre of pycnocline. *Carpenter*, *Balmforth* and *Lawrence* (2010) repositioned the centre of pycnocline, and when the shear centre offset the pycnocline, a higher Richardson number appeared because of a larger buoyancy frequency.

**Question 11**

P 6 line 5. It is misleading to reference mode 1 work here as the initial condition (set up behind the gate) is different and in fact it is the initial condition that is crucial in generating a mode-2 wave (as opposed to mode-1). It would be more appropriate to reference just Brandt & Shipley along with mode-2 papers such as Olsthoorn et al 2013 and/or Deepwell & Statsna 2016, and/or Statsna et al 2015.

**Response:**

Accepted. The citation has been modified.

The related descriptions were added to the revision as follows (see also Page 7, Lines 3 – 5 in the main text):

"*A rank-ordered mode-2 ISW train was generated by the "lock-release" method (Brandt and Shipley, 2014; Olsthoorn* et al*., 2013; Deepwell and Stastna 2016; Stastna* et al*., 2015).*"

**Question 12**

Figure 3: The authors have chosen to offset the shear centre downward of the pycnocline. Do they expect to see similar results (but symmetrically reversed) if it were to be offset in the upward direction? Presumably as the pycnocline centre is at mid-depth. What would happen however if the pycnocline centre were not at mid depth? Also the authors have chosen the shear such that the current in the top layer is in the same direction as the wave - this is similar to the overtaking cases in the work by Stastna et al 2015 and some comparison with that work should be given. Do the authors expect to see the same or different dynamics if the polarity of the shear current is reversed?

**Response:**

We improved and enriched the configuration of the experiment to generalize our

research on the evolution of mode-2 ISWs in shear currents. We also added a comparison to *Stastna* et al. (2015) in the revision.

When the background shear current is shifted upward, the amplitude of the oscillating tails and amplitude-modulated wave packet show nearly symmetrical variation trends compared to those with downward shifts. The amplitude of the forward-propagating long wave was insensitive to the offset of shear current (Figure 1 (a) and (b)). The energy losses of mode-2 ISW are also negatively proportional to the asymmetry parameter Δ in both upward and downward conditions (Figure 2(a) and (b)).

A polarity-reversal background shear current only reversed the polarity of the amplitude-modulated wave packet, oscillating tail and forward-propagating long wave.

Second, as we described in the response to Question 9, an asymmetric stratification can amplify existing instability and induce asymmetrical instability (*Olsthoorn* et al., 2010) as well as additional energy loss in mode-2 ISWs (*Carr* et al., 2015). Therefore, a higher energy loss rate is expected in the asymmetrical stratification.

[Figure]

**Figure 1.** The summarized results of the amplitudes of the forward-propagating long wave (denoted by 'fp'), oscillating tail (denoted by 'ot') and amplitude-modulated wave packet (denoted by 'am') with the presence of (a) upward offset background shear currents at 30 T and (b) downward offset background shear currents at 30 T.

[Figure]

**Figure 2.** The summarized results of the energy loss of the mode-2 ISW at 30 T with the presence of (a) upward

offset background shear currents and (b) downward offset background shear currents.

The related descriptions were added to the revision as follows (see also Page 12, Lines 6 – 12; Page 15, Line 27 – Page 16, Line 3 ;Page 26, Line 4 – Page 27, Line 2 in the main text):

"*In case P1, the polarity-reversal background shear current was initialized in the model. Both in case P1 and the control experiment, the forward-propagating long wave, oscillating tail and amplitude-modulated wave packet could be clearly observed. The properties of the wave structures in the two cases were compared and no significant difference were found. The polarity of the forward-propagating long wave, oscillating tail and amplitude-modulated wave packet are reversed in case P1. This result indicates that the polarity of those shear-induced wave structures is closely related to the polarity of the background shear current.*"

"*A similar variation trend could be found in the upward offset cases (Fig .9 (c)). The amplitudes of the oscillating tail and amplitude-modulated wave packet decreased monotonically as the shear current was offset upward. The forward-propagating long wave was barely affected by Δ and remained constant at approximately 0.2 m in all offset cases.*"

"*Stastna* et al. *(2015) investigated the mode-2 ISW interaction with mode-1 ISW at the same scale. The authors concluded that the shear current is vital, while the*

*deformation of the pycnocline only slightly altered the structure of the mode-2 ISW. For our results, we focused on the effect of shear current, which could be induced by baroclinic eddies, baroclinic tides or wind. We found a high energy loss rate during the modulation of mode-2 ISWs in the presence of background shear current, which is coincident with conclusion given by Stastna* et al. *(2015).*"

**Question 13**

P. 7 line 6 - it'd be useful if $c_p$ were given and/or c presented in non-dimensional form.

**Response:**

The nondimensional form has been used in the revised paper (see also Page 8, Line 8).

**Question 14**

Figure 4 caption: (a) "wave form" is this temperature ? What quantity and scale is the colour bar ?

**Response:**

The 'wave form' is the density field of the initial mode-2 ISW. The caption has been modified, and the quantity and scale were added.

**Question 15**

Page 9 text and figures - it is difficult to see the forward propagating waves - can this be improved ?

**Response:**

This figure has been re-plotted, and the corresponding description has been revised.

**Question 16**

Page 13 line 8 - what are $x_r$ and $x_l$ taken to be though ?

**Response:**

The definitions of $x_l$ and $x_r$ have been clarified. $x_l$ and $x_r$ are denoted as the left and right boundaries, respectively, where the available potential energy flux equals zero (*Lamb*, 2010).

The related descriptions were added to the revision as follows (see also Page 17, Lines 6 – 8 in the main text):

"$x_r$ *and* $x_l$ *are the boundary locations of the integration region, and x satisfies* $x_l \leq x \leq x_r$. *During the calculation of the wave energy,* $x_r$ *and* $x_l$ *are denoted as the left and right boundaries, respectively, where the available potential energy flux equals zero (Lamb, 2010)*"

**Question 17**

Page 14 line 19 - confusing grammar suggests mode-1 are also short lived

**Response:**

Improved.

The related descriptions were added to the revision as follows (see also Page 19, Lines 9 – 11 in the main text):

"*Modulated by the background shear current, the mode-2 ISW exhibits a highly dissipated nature, and the high energy loss rate is comparable to that of the longer mode-1 ISW (Lamb and Farmer, 2011; Shroyer et al., 2010).*"

**Question 18**

Page 17 line 22 - are the authors referring to the field here or their simulations?

**Response:**

The references have been included to support our finding.

"*The superposition of an initially stable shear current and the mode-2 ISW induced a low Ri region with a minimum value of less than 0.01 in our simulation, indicating a possible development of shear instability (Barad and Fringer, 2010).*"

**Question 19**

Figs 13 and 14 and related discussion. If shear instability is present would you not expect to see overturning isopycnals ?

**Response:**

A zoom-in plot of the density contour at 2.8 T for the control experiment (case O5, Δ = 0) is provided to show the overturning process (Figure 3). The region of interest corresponds to Figure 16 (b) in the revised manuscript, which is accompanied by low Ri values. We included this comment and plot in the revision.

[Figure]

**Figure 3.** The density contour plot at 2.8 T for the control experiment (case O5).

The related descriptions were added to the revision as follows (see also Page 24, Lines 23 – 24 in the main text):

"*The overturning process in the isopycnal could also be observed in the corresponding low Ri region (Fig. 17).*"

**Question 20**

Page 19 Line 6 onward. Nice discussion which makes things a lot clearer for the reader, may be this should be given much earlier in the paper.

**Response:**

Thank you for your constructive suggestion. We have polished the structure of this paragraph and improved the description. It has been repositioned earlier in the revision. (See also Page 23, Lines 1 - 16 in the main text)

**Question 21**

Page 20 line 2. This is not clear - there was no background shear in the papers cited in line 1. What do the authors mean here by shear?

**Response:**

This sentence has been revised, and some closely related works have been cited.

The related descriptions were added to the revision as follows (see also Page 24, Lines 2 – 4 in the main text):

"*An oscillating tail induced by shear was also observed in similar studies (Carr* et al.*, 2011, Stamp and Jacka, 1995). The generation of this feature could be related to the shear, and the tail was sustained by continuous energy input.*"

**Reference**

Carpenter J R, Balmforth N J, Lawrence G A. Identifying unstable modes in stratified shear layers[J]. Physics of Fluids, 22(5): 054104, 2010.

Carr M, Davies P A, Hoebers R P. Experiments on the structure and stability of mode-2 internal solitary-like waves propagating on an offset pycnocline[J]. Physics of Fluids, 2015, 27(4): 046602.

Deepwell D, Stastna M. Mass transport by mode-2 internal solitary-like waves[J]. Physics of Fluids, 28(5): 056606, 2016.

Deepwell D, Stastna M, Carr M, et al. Interaction of a mode-2 internal solitary wave with narrow isolated topography[J]. Physics of Fluids, 29(7): 076601, 2017.

Lamb K G. Energetics of internal solitary waves in a background sheared current[J]. Nonlinear Processes in Geophysics, 17(5): 553, 2010.

Lamb K G. Internal wave breaking and dissipation mechanisms on the continental slope/shelf[J]. Annual Review of Fluid Mechanics, 46: 231-254, 2014.

Lamb K G, Farmer D. Instabilities in an internal solitary-like wave on the Oregon shelf[J]. Journal of Physical Oceanography, 41(1): 67-87, 2011.

Olsthoorn J, Baglaenko A, Stastna M. Analysis of asymmetries in propagating mode-2 waves[J]. Nonlinear Processes in Geophysics, 20(1): 59-69, 2013.

Stastna M, Olsthoorn J, Baglaenko A, et al. Strong mode-mode interactions in internal solitary-like waves[J]. Physics of Fluids, 27(4): 046604, 2015.

Terletska K, Jung K T, Talipova T, et al. Internal breather-like wave generation by the second mode solitary wave interaction with a step[J]. Physics of Fluids, 28(11): 116602, 2016.

Yang Y J, Fang Y C, Tang T Y, et al. Convex and concave types of second baroclinic mode internal solitary waves[J]. Nonlinear Processes in Geophysics, 17(6): 605, 2010.

---

## Author Comment (AC3) · 24 Apr 2018

**Response to Reviewer 3:**

**General comment:**

The evolution of the mode-2 internal solitary waves in fluid with stratification in density and shear flow is studied numerically with use MITgcm software. It is demonstrated that initial solitary wave disturbance splits on several wave groups: forwarding-propagated long waves, amplitude-modulated wave packet (which is identified as breather) and oscillating tail behind of soliton. Such groups attenuate with distance due to transfer processes, viscosity and shear instability. Interesting moment of this study is also the difference in the location of pycnocline and shear flow. Obtained results are important for understanding processes related with mode-2 internal waves and I may recommend publishing given paper.

**Response:**

Thank you very much for your constructive and helpful comments, which are highly valuable for us in improving the presentation and quality of the manuscript. We have carefully read the comments and made substantial revisions accordingly. We hope you find these revisions acceptable, and we greatly appreciate your suggestions and comments. We highlight the main revisions in the manuscript, and a point-to-point response is provided below.

**Question 1**

In fact, the difference between oscillating tail and amplitude-modulated packet (breather) is not clear visible. For instance, the modal structure is discussed on Fig. 10, but it will be more useful to see modal structure on waves for different times. The first mode contributes in energy mainly, but what is a difference in amplitudes of modes in the pycnocline?

**Response:**

The related figures have been improved to clearly demonstrate the structures of the oscillating tail and amplitude-modulated wave packet. The amplitude-modulated

wave packet appeared at the end of oscillating tail as steady-state envelopes *(Terletska et al., 2016)*, and it could be clearly observed at the end of the oscillating tail (Figure 1). We have also added a plot to show the modal structures of waves for different times accordingly (Figure 2 and 3). Mode-1 shows the same depression or elevation on both sides of the pycnocline, while mode-2 exhibits a "concave" or "convex" nature, causing elevated and depressed amplitudes on both sides of the pycnocline simultaneously.

[Figure]

**Figure 1.** The evolution process of mode-2 ISW in case O5 at 14 T, where the 'ot' and 'am' denoted the oscillating tail and amplitude-modulated wave packet, respectively (see also FIigure 4(d) in the manuscript).

[Figure]

**Figure 2.** Percent contributions of mode-1 and mode-2 to the total kinetic energy in control experiment (case O5 with Δ = 0) from 0 to 30 T at the (a) front and (b) rear of the mode-2 ISW, the dash lines indicate the cross section where the model structures on waves shown in Figure 2.

[Figure]

**Figure 3.** The model structures on mode-1 (red solid line) and mode-2 (blue solid line) waves in front of the mode-2 ISW at (a) 10 T, (b) 22 T and in rear of the mode-2 ISW at (c) 4 T, (d) 10 T, (e) 16 T, (f) 22 T and (g) 28 T for control experiment (case O5).

The related descriptions were added to the revision as follows (see also Page 18, Lines 13 – 18 in the main text):

"*The model structures of forward-propagating long wave for different times show its mode-1 nature was stable during the evolution of mode-2 ISW in the background shear current (Fig. 12 (a) and (b)). In the rear of the mode-2 ISW, the model structures of trailing waves transformed slightly with the time (Fig. 12 (c) – (g)), it shows the different in vertical structures between amplitude-modulated wave packet (Fig. 12 (c) and (d)) and oscillating tail (Fig. 12 (e), (f) and (g))*"

**Question 2**

What is an origin of the forward-propagating long waves? Are they generated in the initial time only?

**Response:**

The forward-propagating long wave was generated by the collapse of mixing induced by shear instability. This feature was generated persistently accordingly to the Hovmöller plot (Figure 4).

[Figure]

**Figure 4.** Hovmöller plot of horizontal velocity without background shear current at the surface. The mode-2 ISW, forward-propagating long wave, oscillating tail and amplitude-modulated wave packet are denoted by 'm2', 'fp', 'ot' and 'am', respectively. The color bar ranges from -0.034 to 0.034 m/s.

The related descriptions were added to the revision as follows (see also Page 11, Lines 1 - 3; Page 24, Lines 6 - 8 in the main text):

"*A Hovmöller plot (Fig. 6) of horizontal velocity without the background shear current at the surface was plotted (Lamb et al., 2014). The forward-propagating long wave, oscillating tail and amplitude-modulated wave packet were found to propagate persistently.*"

"*Forward-propagating long waves were also observed by Yuan et al. (2018), who found that some small but significant long wavelength mode-1 waves appeared ahead of mode-2 ISWs. The forward-propagating long wave was generated by the collapse of mixing induced by shear instability.*"

**Question 3**

Usually oscillatory tail is generated behind solitary wave due to dissipation, and its

energy is increased when soliton energy is decreased. Is it observed in numerical simulations?

**Response:**

The total energy of the mode-2 ISW and the oscillating tail in numerical simulation was calculated (Figure 5). After the generation of the oscillating tail, its energy increased, while the total energy of the mode-2 ISW decreased.

[Figure]

**Figure 5.** The total energy of the mode-2 ISW and oscillating tail in different periods.

**Technical comments:**

**Question 1**

Axis z is directed up. Looking on formulas (1) and (2) $z = 0$ corresponds to the fluid surface, but on Fig. 3 – to the fluid bottom.

**Response:**

Corrected.

**Question 2**

Fig. 6. There is no vorticity scale.

**Response:**

Added.

**Reference**

Terletska K, Jung K T, Talipova T, et al. Internal breather-like wave generation by the second mode solitary wave interaction with a step[J]. Physics of Fluids, 28(11): 116602, 2016.

---

## Editor Comment (EC1) · M. Stastna (Editor) · 26 Apr 2018

Dear Authors, please ensure that your revised manuscript is available on the Copernicus site for the reviewers to examine. One reviewer contacted me yesterday to say that he thought that the version posted is the original.

Thank you,

Marek Stastna

---

## Author Comment (AC4) · 29 Apr 2018

**Response letter and revised manuscript for**

"The evolution of mode-2 internal solitary waves modulated by

**background shear currents"**

**Peiwen Zhang, Zhenhua Xu, Qun Li, Baoshu Yin, Yijun Hou, Antony K. Liu**

**April 29, 2018**

Dear Prof. Stastna,

On behalf of my co-authors, we greatly appreciate you and the reviewers for the constructive comments and suggestions as well as your time and efforts on processing and reviewing our manuscript. We have carefully read and consider the comments and make substantial revisions accordingly. A point by point response and the revised manuscript are provided below. In the manuscript, the main revisions were highlighted for reviewing, and a clean version of revised manuscript was uploaded separately as a supplement. We hope you find these revisions acceptable.

Some major revisions were made to address the reviewers' concern.

(1) Considering the reviewers' suggestions and comments, several important factors were investigated to generalize our research, including the magnitude, thickness of shear layer, direction, polarity and upward offset of background shear current.

(2) Abstract, results, discussion and summary were reorganized, with latest simulation results added.

(3) We added the definitions of various quantities and improved some descriptions to clarify related claims.

(4) Wrong references, typos, repetition and grammar errors were corrected. Furthermore, we also used the English editing service (NPG) to improve the language. We hope the revised manuscript advanced in readability and understanding.

Additionally, we have remapped some of the figures and improved their resolutions to make them good-quality for reading and publication, and we also have made some minor modifications on the basis of interactive discussions. We hope that you will find the revised manuscript much improved.

Thank you for your attention and consideration of our manuscript.

Yours sincerely

Dr. Zhenhua Xu

Corresponding author: Z. Xu; E-mail: xuzhenhua@qdio.ac.cn

**Response to Reviewer 1:**

**General comments**

The article presents novel numerical results describing the adjustment of a mode-2 ISW due to a background shear flow. The authors vary the center of the background shear and measure the amount of energy lost from the ISW. The authors attempt to demonstrate that this energy is radiated into three different types of waves: a leading mode-1 wave, an oscillating tail, and an amplitude-modulated wave packet. Comparison to the work of Shroyer et al. (2010) is made throughout.

The paper is well structured and contains new results that are applicable to a wide audience. However, substantial work is required to bring the paper up to an international standard. For example, there is missing literature review on mode-1 ISWs in back- ground currents and wave-mean flow interaction, various quantities are not carefully defined, and five cases do not suggest a fully developed study. Detailed suggestions and questions are below. I would like to see the article published, but the authors must address the following concerns.

**Response:**

Thank you very much for your constructive and helpful comments, which were extremely helpful in improving the manuscript. We have carefully read the comments and made substantial revisions accordingly as detailed in the following responses to specific questions. The NPG Language Editing service was contracted to review and polish the revised manuscript before submission. The main revisions are highlighted in the manuscript.

**Question 1**

There are no references to similar studies of mode-1 ISWs in shear flow. Comparisons to lower mode internal waves should be made to better position this article within the literature. Suggestions include: Stastna and Lamb (2002), and Lamb (2010).

**Response:**

2

In response to the reviewer's helpful comments, references related to mode-1 ISWs in shear flow have been added and discussed in the revision. Particularly, we have used the same method as that in *Lamb* (2010) to calculate the ISW energy in the presence of the background current. Some analysis methods in *Stastna and Lamb* (2002) are also used in the present study to investigate the variation of mode-2 wave properties in background shear current.

**The related descriptions were added to the revision as follows (see also Page 2, Line 27 – Page 3, Line 5 in the main text):**

"The evolution of mode-1 ISWs in background shear current was extensively studied (Stastna and Lamb, 2002, Lamb, 2010, Grimshaw et al., 2007, Fructus et al., 2009). Lamb (2010) investigated the energetics of mode-1 ISWs in a background shear current, providing some methods commonly used to calculate the energy under that circumstance. Stastna and Lamb (2002) considered the effects of background current on mode-1 ISWs and discussed the properties of ISWs during the breaking process. In comparison, few works on mode-2 ISW in shear flow have been produced."

**Question 2**

Other than Shroyer et al. (2010), what other references exist for mode-2 ISWs in background currents? Are there none? That seems surprising.

**Response:**

Accordingly, the studies of mode-2 ISWs in the background current have been reviewed and summarized. Because of the difficulty to capture both the mode-2 waves and low-frequency background flows, to our knowledge, *Shroyer* et al. (2010)'s study is the sole one with the observational evidence based on which we can numerically examine the integrated evolution process of mode-2 ISWs in the background shear current.

The related descriptions were added to the revision as follows (see also Page 3, Lines

**5-8; Page 3, Lines 10-13 in the main text):**

"Vlasenko et al. (2010) observed mode-2 ISW followed by short wavelength mode-1 oscillating tail in the Luzon Strait. Liu et al. (2013) investigated the generation and evolution of mode-2 ISW in the South China Sea and concluded that the more dispersive mode-2 ISW might not propagate, evolve and persist for long time on the shelf."

"Shroyer et al. (2010) was the sole one recording an integrated evolution process of mode-2 ISW and found that the leading mode-2 wave quickly deformed and developed a tail of short, small-amplitude mode-1 wave in the presence of background shear current."

**Question 3**

What predictions does theory make? Do the author's results match those of weakly nonlinear theory? It appears that KdV theory will apply and much can be learned by using standard techniques of KdV theory.

**Response:**

The initial properties of mode-2 ISW without the presence of a background shear current in our study is consistent with the prediction by KdV (Korteweg-de Vries) theory (Figure 1) (*Grimshaw* et al., 2007). As introduced by *Maderich* et al (2010), the weakly nonlinear KdV theory is suitable for slowly varying background conditions, but the intensive evolution process is not expected in the KdV framework, and as a result, a numerical modal approach is needed (*Grimshaw* et al., 2010; *Terletkska* et al., 2016; *Yuan* et al., 2018). The recent work by *Yuan* et al., (2018) observed the existence of a long mode-1 wave ahead of the mode-2 ISW during the evolution of mode-2 ISW over variable topography and found that this process cannot be characterized by KdV theory. Therefore, in the present study, we use the full-nonlinear and nonhydrostatic MITgcm model to examine the integrated evolution process of mode-2 ISW in the background shear current.

**Figure 1.** The characteristics of a single mode-2 ISW's (a) density field and (b) vorticity in the absence of background shear current. The white lines in (a) demonstrate the theoretical solution of mode-2 ISW in KdV framework.

**The related descriptions were added to the revision as follows (see also Page 2, Lines 19 - 23 in the main text):**

"Yuan et al. (2018) observed the existence of a long mode-1 wave ahead of mode-2 ISW during the evolution of mode-2 ISW over variable topography and found that this process cannot be characterized by KdV theory. The authors suggested using the MITgcm model, which can solve all modes to investigate the integrated evolution process of mode-2 ISWs in variable background conditions."

**Question 4**

Do five cases provide sufficient information to make generalized claims about ISWs in shear flow? For comparison, Maderich et al. (2017) ran 35 cases of collisions of internal waves.

**Response:**

Thank you for the constructive suggestion. Accordingly, several important factors were investigated to generalize our research in the literature, including the magnitude, thickness of shear layer, direction and symmetric offset of background shear current. The detailed configuration of the numerical simulation is given in Table 1.

**Table 1.** Summary of parameters of variable background shear currents. The depth and thickness of the pycnocline are denoted by  $z_0$  and h. The thickness of the shear layer is denoted by  $h_s$ , and the offset cases are indicated by asymmetry parameter  $\Delta$ . The magnitude of the shear current is denoted by  $U_d$ . Por indicates the polarity of the background shear current, and "+" means a polarity-reversal shear current. The orientation of the background shear

| Case    | $Z_0$ | h  | $h_s$ | Δ    | $U_d$ | Por | Ori |
|---------|-------|----|-------|------|-------|-----|-----|
| 01      | 50    | 10 | 3     | -2   | 0.22  | (-) | (-) |
| O2      | 50    | 10 | 3     | -1.5 | 0.22  | (-) | (-) |
| 03      | 50    | 10 | 3     | -1   | 0.22  | (-) | (-) |
| O4      | 50    | 10 | 3     | -0.5 | 0.22  | (-) | (-) |
| 05      | 50    | 10 | 3     | 0    | 0.22  | (-) | (-) |
| 06      | 50    | 10 | 3     | 0.5  | 0.22  | (-) | (-) |
| 07      | 50    | 10 | 3     | 1    | 0.22  | (-) | (-) |
| 08      | 50    | 10 | 3     | 1.5  | 0.22  | (-) | (-) |
| 09      | 50    | 10 | 3     | 2    | 0.22  | (-) | (-) |
| H1      | 50    | 10 | 1.5   | 0    | 0.22  | (-) | (-) |
| H2 (O5) | 50    | 10 | 3     | 0    | 0.22  | (-) | (-) |
| H3      | 50    | 10 | 4.5   | 0    | 0.22  | (-) | (-) |
| H4      | 50    | 10 | 6     | 0    | 0.22  | (-) | (-) |
| H5      | 50    | 10 | 7.5   | 0    | 0.22  | (-) | (-) |
| U1      | 50    | 10 | 3     | 0    | 0.11  | (-) | (-) |
| U2 (O5) | 50    | 10 | 3     | 0    | 0.22  | (-) | (-) |
| U3      | 50    | 10 | 3     | 0    | 0.33  | (-) | (-) |
| U4      | 50    | 10 | 3     | 0    | 0.44  | (-) | (-) |

current is indicated by Ori, and "+" means an opposing shear current.

| U5 | 50 | 10 | 3 | 0 | 0.55 | (-) | (-) |
|----|----|----|---|---|------|-----|-----|
| D1 | 50 | 10 | 3 | 0 | 0.22 | (-) | (+) |
| P1 | 50 | 10 | 3 | 0 | 0.22 | (+) | (-) |

**The related descriptions were added to the revision as follows (see also Page 3, Lines 17 - 21; Page 5, Lines 2 - 7 in the main text):**

"To reveal the sensitivity of the evolution of mode-2 ISWs to variable parameters of background shear currents, we introduced five sets of experiments (21 experiments in total) to generalize our research, including the magnitude, thickness of the shear layer, direction and offset (centre of the shear layer relative to the centre of the pycnocline) of the background shear current."

"In the sensitive experiments, the magnitude of the shear current is denoted by  $U_d$  and defined as  $|\Delta U|$ . This value was varied from  $0.5U_d$  to  $2.5 U_d$  in the sensitivity test. Similarly, the thickness of the shear layer was varied from  $0.5 h_s$  to  $2.5 h_s$ . In cases D1 and P1, an opposing and polarity-reversal background shear currents were initialized for examination, respectively. We further introduced an asymmetry parameter  $\Delta$  to investigate the evolution of mode-2 ISWs in the offset background shear current (Carpenter et al., 2010)."

**Question 5**

How were the values of the shear chosen? Were they chosen to match flow on the New Jersey shelf? What would happen if the shear was varied in terms of magnitude, or was oriented against the ISW (such that U1 and/or U2 were positive). What about shifting the shear upwards rather than down? What about hs compared to h? Please add some of these cases into the article. Further motivation for the values is required.

**Response:**

In the case O5, which was taken as a control experiment, the values of background

shear current were chosen to match the observations in the New Jersey Shelf (*Shroyer* et al., 2010). The evolution process and calculated energy loss rate for the control experiment were in relative agreement with the field observations.

According to the reviewer's suggestions in questions 4 and 5, we have run more sensitive experiments in the revision (14 additional experiments in total). The main results include the following:

**Orientation**

The background shear current oriented against the mode-2 ISW only slightly affects the amplitude of the forward-propagating long wave and oscillating tail but significantly influences the amplitude of the amplitude-modulated wave packet. The opposing background shear current slightly increased the energy loss of mode-2 ISW during the modulation.

**Magnitude of the shear**

The amplitudes of the forward-propagating long wave and amplitude-modulated wave packet are positively proportional to the magnitude of the background shear current, but the oscillating tail is not very sensitive to the increasing magnitude of the background shear current (Figure 2 (a)). The energy losses of mode-2 ISW are also positively proportional to the magnitude of the background shear current (Figure 3 (a)).

**Thickness of the shear layer**

The amplitudes of all three shear-induced wave structures are negatively proportional to the thickness of the shear layer (Figure 2 (b)). The energy losses of mode-2 ISW are also negatively proportional to the thickness of the shear layer (Figure 3 (b)).

**Offset upward**

When the shear current is offset upward, the variation of the oscillating tail and amplitude-modulated wave packet in amplitude show similar trends to the downward offset cases (Figure 2 (c) and (d)), but the forward-propagating long wave is still insensitive to the offset direction of the shear current (Figure 2 (c) and (d)). The energy losses of mode-2 ISW are also negatively proportional to the asymmetry parameter  $\Delta$  in both upward and downward conditions (Figure 3(c) and (d)).

**Figure 2.** The summarized results of the amplitudes of the forward-propagating long wave (denoted by 'fp'), oscillating tail (denoted by 'ot') and amplitude-modulated wave packet (denoted by 'am') with the presence of (a) varied magnitude of shear currents at 40 T, (b) varied thickness of shear currents at 40 T, (c) upward offset background shear currents at 30 T and (d) downward offset background shear currents at 30 T.

---

## Referee Report (RR1)

**1    General comments**

The authors have improved the presentation and quality of the article, but a few comments remain.

**2    Comments**

1. Figure 15 shows a large difference in the energy loss whether the shear layer has moved up or down. What causes this significant difference? This variation is interesting and warrants discussion.

2. The definition of APE uses $x_l$ and $x_r$. However, these are defined in terms of $\text{APE}_f$, which is defined in terms of APE. This appears circular. Clarity in regards to the APE density used in equation 9 should be made.

3. Please list $U_0$ for each experiment.

4. Pg 10, line 20. How do you know that the oscillating tail appears after the amplitude-modulated wave-packet has propagated away? Could it not generate simultaneously and appear as part of the amplitude-modulated wave?

5. Pg 18, line 17. What do you mean by "far away from the mode-2 ISW"? Are the model structures being calculated at different positions behind the wave at different times? Are they not consistently placed a fixed distance behind the mode-2 ISW?

6. Pg. 20, line 15. It is not clear to me how figure 12 is connected to the energy flux. Did you mean figure 13?

7. Please confirm, but I think the authors want proportional and inversely proportional rather than positively proportional and reverse proportional, respectively. The latter have been used throughout.

---

## Author Response (AR2)

**Response letter and revised manuscript for**

*"The evolution of mode-2 internal solitary waves modulated by*

*background shear currents"*

Peiwen Zhang, Zhenhua Xu, Qun Li, Baoshu Yin, Yijun Hou, Antony K. Liu

May 28, 2018

Dear Prof. Stastna,

On behalf of my co-authors, we greatly appreciate you and the reviewers for the positive and constructive comments, as well as your time and efforts on processing and reviewing our manuscript. We have carefully considered the comments and make substantial revisions to address the editor's and reviewers' concerns. The NPG Language Editing service had been contracted to review and polish the revised manuscript before the submission. A point-by-point response and a marked-up manuscript showing the revisions are provided below, and a clean version of revised manuscript was uploaded separately as a supplement. We are looking forward to hearing from you.

Thank you for your attention and consideration of our manuscript.

Yours sincerely,

Zhenhua Xu

Corresponding author: Z. Xu; E-mail: xuzhenhua@qdio.ac.cn

**Response to Reviewer 1:**

**General comments**

The authors have improved the presentation and quality of the article, but a few comments remain.

**Response:**

Thank you very much for your constructive and valuable comments which are greatly helpful for us to improve the manuscript. We have carefully read the comments and revised the manuscript accordingly. We hope you find these revisions acceptable.

**Question 1**

Figure 15 shows a large difference in the energy loss whether the shear layer has moved up or down. What causes this significant difference? This variation is interesting and warrants discussion.

**Response:**

Thank you for pointing this out. The structures of amplitude and wave-induced current of mode-2 ISW are symmetric in the vertical, but the wave-induced shear is asymmetric (Figure 1). The difference in energy loss when the background shear current moves upward or downward is caused by the asymmetry of wave-induced shear. We further found that when background shear current offset upward, the oscillating tail appeared with larger amplitude and the wave-induced shear had opposite pattern to background current shear. In contrast, the oscillating tail emerged with relatively small amplitude when background shear current offset downward (Figure 10 in the manuscript). The related description has been added in the revision.

[Figure]

**Figure 1.** The vertical structure of current (a) and shear (b) induced by mode-2 ISW, and (c) background shear current in Case O5.

The related descriptions were added to the revision as follows (see also Page 22, Lines 13 – 17 in the main text):

"*We further found that the energy losses of mode-2 ISW in upward conditions were larger compared to the downward conditions. This phenomenon is caused by the asymmetry of wave-induced shear. When the shear layer moves up or down, the background current could weaken the shear in upper layer or strengthen the shear in lower layer, causing the difference in energy losses of mode-2 ISW.*"

**Question 2**

The definition of APE uses $x_l$ and $x_r$. However, these are defined in terms of $APE_f$ , which is defined in terms of APE. This appears circular. Clarity in regards to the APE density used in equation 9 should be made.

**Response:**

We are sorry for our unclear description. In equation (9), "APE" was modified to "$E_{ape}$", which denoted the available potential energy.

The related descriptions were revised as follows (see also Page 17, Lines 18 – 24 in the main text):

"*They are written as:*

$$KE_f = \int_{-H}^{0} uE_{ke}dz,\qquad\qquad(8)$$

$$APE_f = \int_{-H}^{0} uE_{ape}dz,\qquad\qquad(9)$$

$$W = \int_{-H}^{0} up_d dz,\qquad\qquad(10)$$

*where u is the horizontal velocity induced by mode-2 ISWs, $E_{ke}$ and $E_{ape}$ are kinetic energy and available potential energy, and $p_d$ is the pressure perturbation relative to the reference state (Lamb and Nguyen, 2009)."*

**Question 3**

Please list $U_0$ for each experiment.

**Response:**

The mean background velocity of water column ($U_0$) for each experiment has been listed in Table 1 as you suggested.

**Question 4**

Pg. 10, line 20. How do you know that the oscillating tail appears after the amplitude-modulated wave-packet has propagated away? Could it not generate simultaneously and appear as part of the amplitude-modulated wave?

**Response:**

According to the Hovmöller plot (Figure 2), the trace of oscillating tail appears later than the amplitude-modulated wave packet. The oscillating tail is located between the mode-2 ISW and amplitude-modulated wave packet due to its relatively higher propagation speed than the wave packet. Meanwhile, the amplitude-modulated wave packet propagates away with nearly constant total energy, exhibiting no significant decreasing in total energy of the wave packet when it falls behind the oscillating tail.

[Figure]

**Figure 2.** Hovmöller plot of horizontal velocity without background shear current at the surface. The mode-2 ISW, forward-propagating long wave, oscillating tail and amplitude-modulated wave packet are denoted by 'm2', 'fp', 'ot' and 'am', respectively.

The related descriptions were added to the revision as follows (see also Page 10, Lines 20 – 22 in the main text):

"*The traces of forward-propagating long wave and amplitude-modulated wave packet appear at the same time (Fig. 6). The oscillating tail appears later than the amplitude-modulated wave packet, and is located between the mode-2 ISW and amplitude-modulated wave packet (Fig. 6).*"

**Question 5**

Pg 18, line 17. What do you mean by "far away from the mode-2 ISW"? Are the model structures being calculated at different positions behind the wave at different times? Are they not consistently placed a fixed distance behind the mode-2 ISW?

**Response:**

It means the amplitude-modulated wave packet propagates far away from mode-2 ISW after its generation. The cross sections for the calculation of modal structure were placed a fixed distance behind the mode-2 ISW. We modified the description to express more explicitly.

The related descriptions were revised as follows (see also Page 18, Lines 16 – 19 in the main text):

"*In the rear of the mode-2 ISW, the model structures of trailing waves transform slightly with the time (Fig. 12 (c) – (g)), and the wave-induced currents tend to be concentrated around the pycnocline when the amplitude-modulated wave packet propagates far away from the mode-2 ISW.*"

**Question 6**

Pg. 20, line 15. It is not clear to me how figure 12 is connected to the energy flux. Did you mean figure 13?

**Response:**

Thank you for pointing this out. "Figure 12" was corrected to "Figure 13".

**Question 7**

Please confirm, but I think the authors want proportional and inversely proportional rather than positively proportional and reverse proportional, respectively. The latter have been used throughout.

**Response:**

Agreed. "Positively proportional" and "reverse proportional" were modified to "proportional" and "inversely proportional" throughout the manuscript, respectively.

**Response to Reviewer 2:**

**General comments**

The authors have improved the paper and answered all of my comments satisfactorily. In particular, they have illustrated and described the features "forward propagating", "oscillating tail" and "amplitude modulated" much better than in the first draft. The inclusion of the Hovmöller plot clearly shows the distinct behavior of the different features and is a welcome addition. Widening of the parameter space has added further weight to their results and is also welcome. I recommend that the paper be published but I still think the grammar could/should be improved before final acceptance.

**Response:**

We sincerely thank you for your valuable and helpful comments, and the manuscript was revised accordingly. We carefully check the grammar and the NPG Language Editing service has been contracted to review and polish the revised manuscript before the submission. We hope you find these revisions acceptable.

**Question 1**

Page 2 line 22 - KdV give full name for non-specialist readers

**Response:**

Added

**Question 2**

Page 2 last line - remove citation Fructus et al. This paper does not have a background shear.

**Response:**

Thank you for pointing this out. The citation of *Fructus* et al (2009) has been removed.

**Question 3**

Fig 2 - misleading to show $h_{mix}$ as being less than h

**Response:**

Modified accordingly.

**Question 4**

Page 24 line 9 - give more info or remove citation e.g. Carr at al 2011 was for mode-1 waves with no background shear.

**Response:**

Thank you for your kind suggestion. The citation of *Carr* et al (2011) and *Stamp and Jacka* (1995) have been removed.

[revised manuscript text omitted]